# Satellite-derived Constraints on the Effect of Drought Stress on Biogenic Isoprene Emissions in the Southeast US

Yuxuan Wang[1], Nan Lin[1], Wei Li[1], Alex Guenther[2], Joey C. Y. Lam[3], and Amos P. K. Tai[3,4], Mark J. Potosnak[5], Roger Seco[6]

[1] Department of Earth and Atmospheric Sciences, University of Houston, Houston, Texas, USA.

[2] Earth System Science, University of California, Irvine, Irvine, California, USA

[3] Earth and Environmental Sciences Programme, Faculty of Science, The Chinese University of Hong Kong, Hong Kong SAR, China

[4] State Key Laboratory of Agrobiotechnology and Institute of Environment, Energy and Sustainability, The Chinese University of Hong Kong, Hong Kong SAR, China

[5] Environmental Science and Studies, DePaul University, Chicago, IL, USA

[6] Institute of Environmental Assessment and Water Research (IDAEA-CSIC), Carrer Jordi Girona 18-26, 08034 Barcelona, Spain

*Corresponding author:* Yuxuan Wang (ywang246@central.uh.edu)

**Abstract.** While substantial progress has been made to improve our understanding of biogenic isoprene emissions under unstressed conditions, there remain large uncertainties in isoprene emissions under stressed conditions. Here we use the US Drought Monitor (USDM) as a weekly drought severity index and tropospheric columns of formaldehyde (HCHO), the key product of isoprene oxidation, retrieved from the Ozone Monitoring Instrument (OMI) to derive top-down constraints on the response of summertime isoprene emissions to drought stress in the Southeast U.S. (SE US), a region of high isoprene emissions and prone to drought. OMI HCHO column density is found to be 6.7% (mild drought) - 23.3% (severe drought) higher than that in no-drought conditions. A global chemical transport model, GEOS-Chem, with the MEGAN2.1 emission algorithm can simulate this direction of change, but the simulated increases at the corresponding drought levels are 1.1-1.5 times of OMI HCHO, suggesting the need for a drought-stress algorithm in the model. By minimizing the model-to-OMI differences in HCHO to temperature sensitivity under different drought levels, we derived a top-down drought stress factor ($\gamma_{d\_OMI}$) in GEOS-Chem that parameterizes using water stress and temperature. The algorithm led to an 8.6% (mild drought) - 20.7% (severe drought) reduction in isoprene emissions in the SE US relative to the simulation without it. With $\gamma_{d\_OMI}$ the model predicts a non-linear increasing trend in isoprene emissions with drought severity that is consistent with OMI HCHO and a single site's isoprene flux measurements. Compared with a previous drought stress algorithm derived from the latter, the satellite-based drought stress factor performs better in capturing the regional scale drought-isoprene responses as indicated by the close-to-zero mean bias between OMI and simulated HCHO columns under different drought conditions. The drought stress algorithm also reduces the model's high bias in organic aerosols (OA) simulations by 6.60% (mild drought) to 11.71% (severe drought) over the SE US compared to the no-stress simulation. The simulated ozone response to the drought stress factor displays a spatial disparity due to the isoprene suppressing effect on oxidants, with an <1 ppb increase in $O_3$ in high-isoprene regions and a 1-3 ppbv decrease in $O_3$ in low-isoprene regions. This

study demonstrates the unique value of exploiting long-term satellite observations to develop empirical stress
algorithms on biogenic emissions where in situ flux measurements are limited.
**1. Introduction**
Biogenic nonmethane volatile organic compounds (BVOCs) emitted by terrestrial ecosystems are of great importance
to air quality, tropospheric chemistry, and climate due to their effects on atmospheric oxidants and aerosols (Atkinson,
2000; Claeys et al., 2004; Pacifico et al., 2009). The dominant BVOC is isoprene ($CH_2=C(CH3)CH=CH_2$), comprising
70% of the global total BVOC emitted from vegetation (Sindelarova et al., 2014). Isoprene emissions depend on
vegetation/plant type, physiological status, leaf age, and meteorological conditions such as radiation, temperature, and
soil moisture. These relationships provide the basic framework of isoprene emission models that are capable of
coupling with meteorology and the land biosphere, the most widely used being the Model of Emissions of Gases and
Aerosols from Nature (MEGAN) (Guenther et al., 1993, 2006, 2012, 2017). Recent work has shown stressed
conditions - such as drought, heatwaves, and high winds - can induce large changes in isoprene emissions different
from model predictions in the absence of those stress factors (Potosnak et al., 2014; Huang et al., 2015; Kravitz et al.,
2016; Seco et al., 2015; Otu-Larbi et al., 2020; Seco et al., 2022). As stressed conditions are rarely sampled by field
campaigns due to their infrequent and irregular nature and hence poorly constrained, stress impacts on isoprene
emissions are among the least understood aspects in our predictivity of BVOC-chemistry-climate interactions.
A common stress for terrestrial vegetation worldwide is drought, characterized by low precipitation, high temperature,
and low soil moisture (Trenberth et al., 2014). These conditions are primary abiotic stresses that will cause
physiological impacts on plants affecting photosynthesis, stomatal conductance, transpiration, and leaf area. During
short-term or mild droughts, the photosynthetic rate of plants quickly decreases due to limited stomatal conductance,
while isoprene is not immediately impacted because of the availability of stored carbon and because the photosynthetic
electron transport is not inhibited. Isoprene can even increase by several factors due to warm leaf temperatures which
increases isoprene synthase activity (Potosnak et al., 2014; Ferracci et al., 2020). During prolonged or severe drought
stress, after a lag related to photosynthesis reduction, isoprene emission eventually declines because of inadequate
carbon availability. This conceptualized non-monotonic response of isoprene emission to drought has been
demonstrated at the Missouri Ozarks AmeriFlux (MOFLUX) field site in Missouri (Potosnak et al., 2014; Seco et al.,
2015), the only available drought-relevant whole canopy isoprene flux measurements to date, and qualitatively
supported by ambient isoprene concentrations monitored by regional surface networks (Wang et al., 2017). It is
noteworthy that the MOFLUX data covered only two drought events (summer 2011 and summer 2012), while the
surface sites are sparsely distributed with an urban focus. More recently, the isoprene concentration measurements
during the Wytham Isoprene iDirac Oak Tree Measurements (WIsDOM) campaign showed that isoprene was up to
four times higher than normal in responses to a combined heatwave and drought episode (June-October 2018) over a
mid-latitude temperate forest in the UK (Ferracci et al., 2020; Otu-Larbi et al., 2020), which supports the enhanced
isoprene emissions at the MOFLUX site under mild droughts. However, these observations offer only limited
constraints on drought stress impacts on isoprene emissions.
With wide spatiotemporal coverage, satellite provides arguably the best platform to capture drought development and
impacts. Satellite observations of tropospheric formaldehyde (HCHO) columns have been used as a proxy of isoprene
emissions for more than a decade (Abbot et al., 2003; Palmer et al., 2003), as HCHO is formed promptly and in high
yield from isoprene oxidation (Sprengnether et al., 2002). Previous applications of satellite HCHO products provided
"top-down" estimates on seasonality, magnitude, spatial distribution, and interannual variability of isoprene emissions
globally and regionally (e.g., Marais et al., 2016; Kaiser et al., 2018; Stavrakou et al., 2018). While most of these
studies focused on *unstressed* conditions, recent efforts have shown that satellite HCHO registered drought signals on
a monthly scale (Zheng et al., 2017; Naimark et al., 2021; Li et al., 2022; Opacka et al., 2022). These signals are yet
to be exploited to constrain isoprene response to drought.
The present study aims at improving the current quantification of satellite HCHO response to drought by accounting
for sub-monthly variability of drought severity. We use a weekly time scale, the finest temporal scale of drought
indices available, and separate five levels of drought severity defined by the US Drought Monitor. By comparison,
previous investigations used binary classification (drought or not) on a monthly time scale. Our improvement in scale
is expected to better capture the nonlinear response of isoprene emissions to drought severity as described above. The
study region is the Southeast United States (SE US), which has large isoprene emissions due to substantial forest
coverage and is also prone to drought due to large interannual variability in precipitation (Seager et al., 2009). In
addition, the MOFLUX site is located in the SE US, which will allow us to evaluate if satellite-derived drought
responses of HCHO are consistent with those from isoprene flux measurements at MOFLUX. Finally, we use these
HCHO signals in conjunction with models to identify the model gaps in predicting isoprene responses to drought.
**2. Data and Method**
**2.1 Drought index**
There are many types of drought indices focusing on different factors, including precipitation, temperature,
evaporation, runoff, and the impact of drought on ecosystems and vegetation (Palmer, 1965; McKee et al., 1993;
Guttman, 1999; Vicente-Serrano et al., 2010; Chang et al., 2018). Drought indices also differ by time scale. As drought
by definition is a prolonged period of water deficit, the shortest time scale of drought is weekly. Here we chose the
United States Drought Monitor (USDM) drought index to identify drought periods. USDM's weekly timescale and
multiple drought severity levels (Svoboda et al., 2002) provide a better delineation of drought variability than the
monthly or seasonal scale used in the previous analysis of drought signals in HCHO and isoprene (Wang et al., 2017;
Naimark et al., 2021).

(a) USDM released on July 12,2012    (b) USDM time series for SE US

☐ N0 Wet and Normal    ■ D0 Abnormal Drought    ■ D1 Moderate Drought
■ D2 Severe Drought    ■ D3 Extreme Drought    ■ D4 Exceptional Drought

**Figure 1. (a) Drought distribution for the second week of July 2012 based on USDM. The black star indicates the location of MOFLUX site. (b) Time series of drought frequency in the study area (black box in Figure 1a) for JJA from 2005 to 2017. N0 (white) for wet and normal, D0 (light yellow) for abnormal drought, D1 (yellow) for moderate drought, D2 (orange) for severe drought, D3 (red) for extreme drought, and D4 (brown) for exceptional drought.**

The USDM is a composite drought index based on six key physical indicators including the Palmer Drought Severity Index (PDSI, Palmer, 1965), CPC Soil Moisture Model Percentiles (Huang et al., 1996), U.S. Geological Survey (USGS) Daily Streamflow Percentiles (http://water.usgs.gov.waterwatch/), Percent of Normal Precipitation (Willeke et al., 1994), Standardized Precipitation Index (SPI, McKee et al., 1993), and remotely sensed Satellite Vegetation Health Index (Kogan, 1995). Opinions of local experts are also considered (Svoboda et al., 2002). The USDM website (https://droughtmonitor.unl.edu/) provides weekly ArcGIS shapefiles of the polygons covering the whole US under five drought levels: D0 for abnormal drought, D1 for moderate drought, D2 for severe drought, D3 for extreme drought, and D4 for exceptional drought. We used the method of Chen et al. (2019) to rasterize and convert USDM shapefiles to $0.5° \times 0.5°$ gridded indices with -1 indicating non-drought (N0) and 0-4 for D0-D4 drought, respectively. **Figure 1a** displays the spatial distribution of gridded USDM indices for the second week of July 2012, which clearly depicts the extent and severity of the infamous 2012 Great Plains drought (Hoerling et al., 2014). **Figure 1b** shows the weekly time series of USDM indices averaged over SE US (75–100∘W, 25–40∘N, black box in Figure 1a) for the summer months (June, July, August; JJA) of 2005 -2017, our study period. During this period, abnormal drought (D0) appeared every summer, while extreme and exceptional drought (D3-D4) were mainly concentrated in 2006-2008 and 2010-2012. This pattern is consistent with the long-term drought statistics from other drought indices such as SPEI and PDSI (Svoboda et al., 2015).

**2.2 OMI HCHO and NO₂ product**

We used the Ozone Monitoring Instrument (OMI) v003 level 3 tropospheric formaldehyde (HCHO) column density (OMHCHOd) as described by Chance (2019). OMI was launched on NASA's Aura satellite in 2004 and has since provided daily global measurements of ozone ($O_3$) and its precursors with a nadir spatial resolution of $24 \times 13$ $km^2$. Since January 2009, OMI has been suffering from a major row anomaly. OMHCHOd data processing explored all level 2 OMHCHO observations to filter out pixels with bad formaldehyde retrievals, high cloud fractions (>30%), high SZA (>70°), and pixels affected by OMI's row anomaly (Chance, 2019). The spatial resolution is $0.1° \times 0.1°$. Zhu et al. (2016) verified the OMHCHOd data using high-precision HCHO aircraft observations obtained during

NASA SEAC4RS activities in SE US from August to September 2013. They showed that OMI retrievals have accurate
spatial and temporal distribution but were biased low by 37% relative to the aircraft. We corrected this underestimation
by applying a uniform and constant factor of 1.5 to the OMHCHOd data, as did by Shen et al. (2019) in their long-
term analysis of OMI HCHO. **Figure 2a** presents the corrected OMHCHOd for the SE US averaged over JJA 2005-
2017, where higher levels of HCHO are clearly seen over forested regions in Missouri, Georgia, Arkansas, and Texas.
OMHCHOd values shown hereafter are those with the correction factor applied. Although it is not known if the
correction factor has temporal spatial variations during our study period, its application produced a good match
between OMI and simulated HCHO columns under non-drought (N0) conditions (Figure 2c). To examine the
concurrent changes of nitrogen oxides ($NO_x = NO_2 + NO$) under droughts, we also used the level 3 tropospheric
column of $NO_2$ from OMI during the same period (Nickolay et al., 2019).

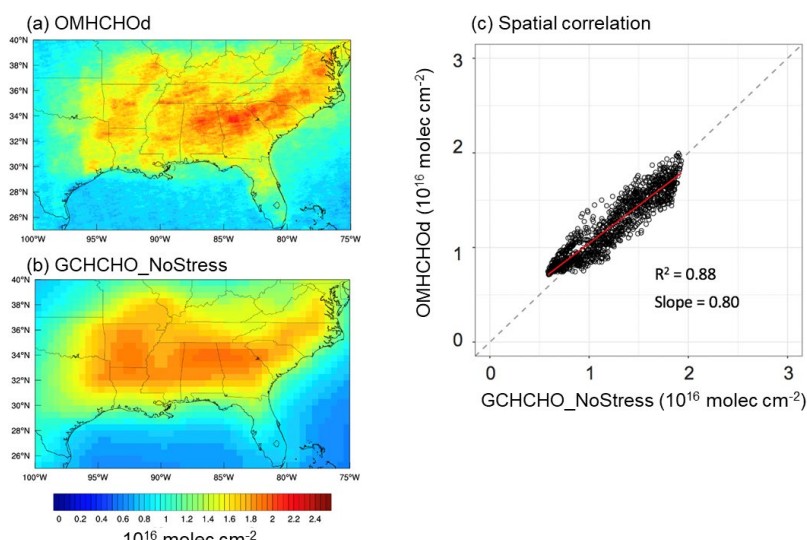

**Figure 2. Mean 2005–2017 HCHO columns for June – August over the SE US of (a) OMI observation (OMHCHOd) and**
**(b) GEOS-Chem simulation (GCHCHO_NoStress). (c) Scatterplot of spatial correlation between the two. The dashed line**
**indicates the 1:1 agreement.**
**2.3 GEOS-Chem chemical transport model**
We used the long-term simulation of the nested-grid GEOS-Chem global chemical transport model (version 12-02,
http://www.geos-chem.org) to obtain daily mean results of modeled formaldehyde columns and isoprene emissions
for North America during JJA 2005 – 2017. The simulation was driven by the Modern-Era Retrospective analysis for
Research and Applications, Version 2 (MERRA-2) meteorological data from NASA's Global Modeling and
Assimilation Office (GMAO) with a horizontal resolution at 0.5° × 0.625°. Biogenic emissions were calculated using
MEGAN2.1, which is the prevailing version of MEGAN implemented in most chemical and climate models.
MEGAN2.1 has a soil dependence algorithm whose parameterization is based on plant wilting point threshold and
soil moisture (Guenther et al., 2017). However, this factor is disabled in GEOS-Chem as in many other CTMs due to
the unavailability of the required driving variables, such as wilting point and soil moisture, which cannot be simulated
well in most models (Trugman et al., 2018). Thus, outputs from the standard GEOS-Chem simulations do not have
drought effects on isoprene emissions and these outputs are referred to as NoStress_GC. Anthropogenic emissions
over North America were from the 2011 National Emissions Inventory (NEI2011, http://www.epa.gov/air-emissions-
inventories) for the United States, with historical scale factors applied to each simulated year. Open fire emissions
were from GFED4 (Giglio et al., 2013) for 2005–2017.
To better match with OMI overpassing time, model HCHO outputs at 13:30 local time were sampled
(GCHCHO_NoStress). **Figure 2b** shows GCHCHO_NoStress averaged over the same domain and period as
OMHCHOd in **Figure 2a**. The scatter plot (**Figure 2c**) shows a good spatial correlation between the two ($R^2 = 0.88$).
This correlation is consistent with other studies comparing GEOS-Chem and OMI HCHO columns in SE US during
non-drought periods (Kaiser et al., 2018).
**2.4 Observations of ozone, organic aerosol, LAI, and isoprene flux**
To evaluate how the drought stress factor changes the simulations of surface $O_3$ and organic aerosol (OA), we adopted
the gridded ($1° \times 1°$) hourly $O_3$ observations created by Schnell et al. (2014) using the modified inverse distance
weighting method. The dataset aggregates several networks of $O_3$ measurements including the US Environmental
Protection Agency's (EPA) Air Quality System (AQS), Clean Air Status and Trends Network (CASTNET), and
Environment Canada's National Air Pollution Surveillance Program (NAPS). Following the same method, we created
a gridded organic aerosol (OA) dataset using the organic carbon (OC) observations from the Interagency Monitoring
of Protected Visual Environments (IMPROVE) network. A factor of 2.1 was used to convert OC to OA as suggested
by other studies (Pye et al., 2017; Schroder et al., 2018). To examine the changes of leaf area index (LAI) under
droughts, the MODerate resolution Imaging Spectroradiometer (MODIS) Collection 5 LAI products reprocessed by
Yuan et al. (2011) with a resolution of $0.25° \times 0.25°$ was used. These three datasets were further remapped through
bilinear interpolation to match the spatial resolution of the USDM. The isoprene flux measurements at the MOFLUX
site during 2012 May-September were used to derive a site-based drought stress algorithm. The site is located in the
Ozarks region of central Missouri (38.74°N, 92.20°W, black star in Figure 1a). It is surrounded by a deciduous forest
dominated by isoprene-emitting white and red oak species. The dataset is widely used to investigate isoprene emissions
response to droughts (Potosnak et al., 2014; Seco et al., 2015; Jiang et al., 2018; Opacka et al., 2022).
**3. Observational Evidence of Drought Stress on Isoprene Emissions**
**3.1 Changes of HCHO column densities with drought**
To reveal drought responses of HCHO, we sampled weekly-mean HCHO columns onto the gridded spatial and
temporal locations of each USDM category and generated average HCHO distributions at each drought level over the
SE US. The outputs are shown in **Figure 3a** for OMI and **3b** for NoStress_GC, respectively. The processing of weekly-
mean HCHO data corresponds to the timing of USDM: a whole week includes Wednesday of the previous week to
Tuesday of the present week. There are 12 consecutive weeks from June to August in each year of 2005-2017, giving
a total of 156 weeks' gridded HCHO data to be assigned to individual USDM categories by week and location. **Figure**
**3d** shows the number of weeks underlying the gridded averages of HCHO for each USDM category. As severe
droughts are less frequent than mild droughts, some locations in SE US did not experience D2-D4 droughts during the
study period and hence are shown as white in **Figure 3**.

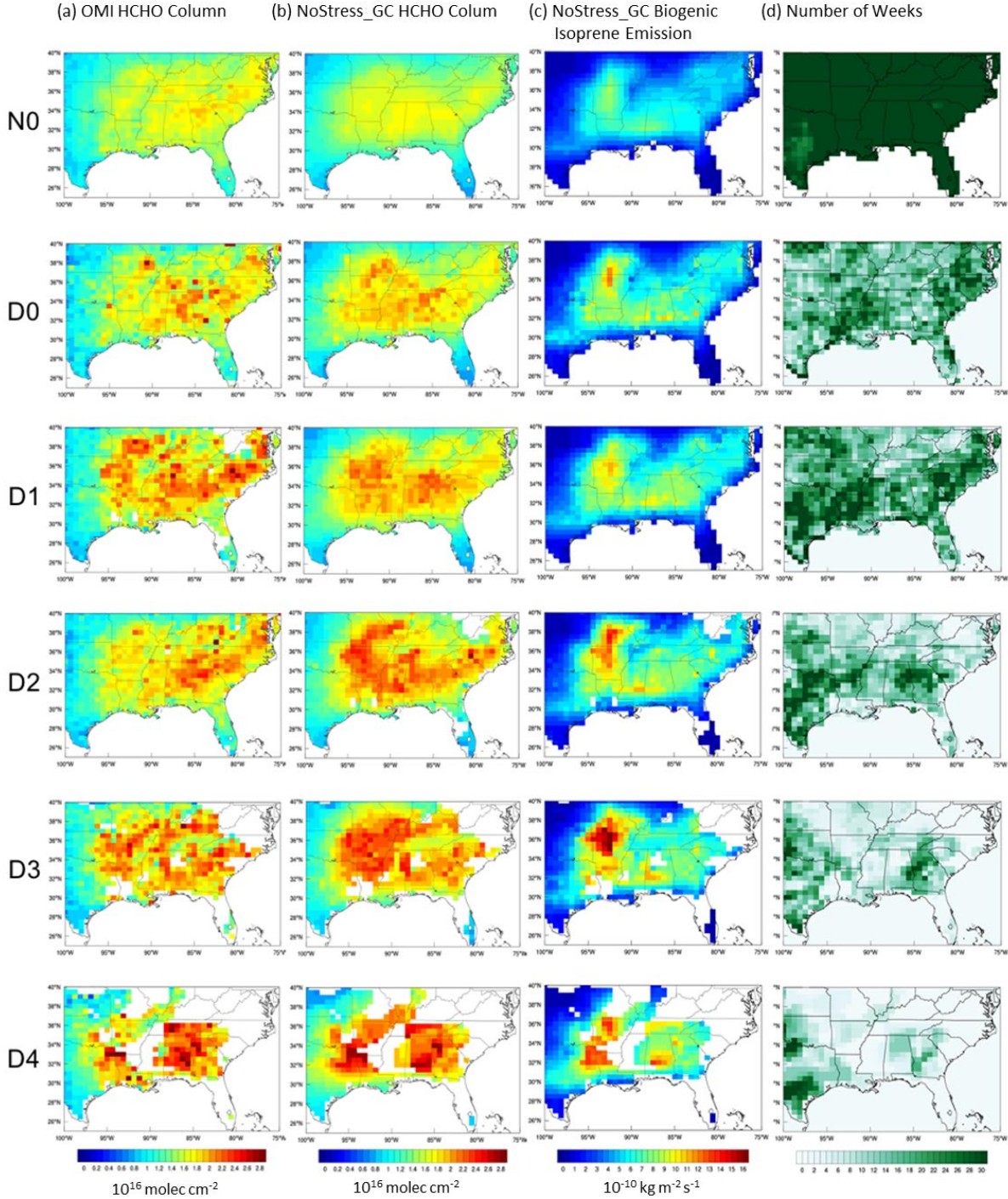


**Figure 3. The mean spatial distributions of (a) OMI HCHO column density; (b) NoStress_GC HCHO column density, (c)**
**NoStress_GC isoprene emissions, and (d) the number of weeks during JJA 2005 to 2017 in the southeast US under different**
**USDM drought levels (N0, D0-D4).**
OMI HCHO column density increases with increasing drought severity in almost all locations in the SE US (**Figure**
**3a**). Relative to no-drought condition (N0), the mean HCHO column from OMI is 6.7%, 12.6%, 16.5%, 21.2%, and
23.2% higher under D0 - D4 drought in the entire SE US, respectively. These HCHO changes are statistically
significant at a 95% confident interval, indicating that the OMI HCHO products contain significant drought signals.
The increasing rate of OMI HCHO with USDM is not linear, faster under mild droughts (D0-D2) and flattening under
more severe droughts (D2-D4). This is qualitatively consistent with the conceptualized model of the nonlinear
response of isoprene emissions to drought described before (Potosnak et al., 2014).
Model HCHO column density also increases with increasing drought severity (**Figure 3b**). GCHCHO_NoStress is
9.90%, 15.1%, 19.5%, 21.8%, and 29.1% higher under D0-D4 drought than that of N0, respectively. These increases
are 1.1-1.5 times those of OMI under all drought levels. The model comparison against OMI HCHO also changes
with drought severity. GCHCHO_NoStress has a minimal bias ($0.05 \times 10^{16}$ molec cm$^{-2}$) under N0. As drought severity
increases, the mean bias over the entire SE US increases to $0.10 \times 10^{16}$ molec cm$^{-2}$, $0.09 \times 10^{16}$ molec cm$^{-2}$, $0.11 \times$
$10^{16}$ molec cm$^{-2}$, $0.08 \times 10^{16}$ molec cm$^{-2}$, and $0.15 \times 10^{16}$ molec cm$^{-2}$ under D0 - D4 levels, respectively. The spatial
correlation between OMI and NoStress_GC degrades with USDM, with $R^2$ being smaller than 0.65 under D0 - D4
levels compared to $R^2$ of 0.70 under N0. Worsening model performance with increasing drought severity suggests the
model lacks a process that changes with drought. As isoprene accounts for more than 80% of the contribution of non-
methane VOCs to the HCHO column in the southeast US(Palmer et al., 2003; Millet et al., 2006), the missing process
is most likely drought-induced changes in isoprene emissions.

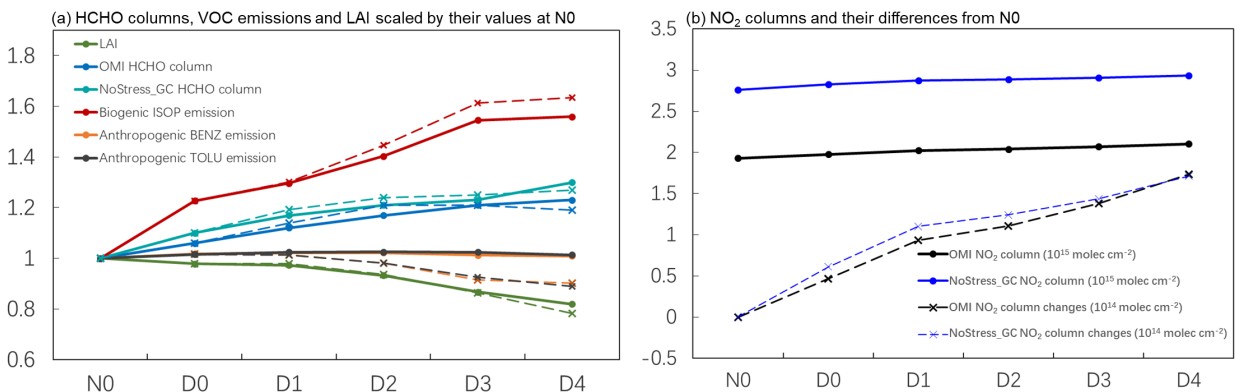


**Figure 4. (a) Relative changes of regional-mean OMI HCHO column, NoStress_GC simulated HCHO colum, isoprene**
**emissions, anthropogenic benzene emission, anthropogenic toluene emission, and MODIS leaf area index (LAI) under**
**different drought levels in the southeast US. All data are scaled to their respective values at N0. The dotted lines are the**
**arithmetic mean of all grids, and the solid lines are the corrected mean excluding the missing area. (b) Regional-mean**
**tropospheric NO₂ columns from OMI and NoStress_GC (solid lines), and their respective changes from non-drought (N0)**
**conditions (dashed lines). Note the different scales between the solid and dashed lines. The calculation is based on the grids**
**with the presence of all USDM levels.**
**Figure 4a** displays the relative changes in the regional mean HCHO column from OMI and NoStress_GC, emissions
of isoprene and select anthropogenic VOCs from NoStress_GC, and MODIS LAI as a function of USDM indices,
each scaled by its respective value at N0. The dotted line is the arithmetic mean of all available grids under each
dryness category, and the solid line is the mean for those grids with valid data in all dryness categories (i.e., removing
white areas shown in Figure 3). In either calculation, NoStress_GC overestimates the relative increase of HCHO under
D0-D4 by 10-50% compared to OMI. After correcting for no data areas at D2-D4, isoprene emissions in NoStress_GC
are 22.7%, 29.6%, 40.3%, 54.5%, and 56.0% higher in D0-D4 than N0. Note that LAI is observed to decrease by 5-
10% per USDM level (**Figure 4a**), which makes the predicted increase of isoprene emissions with drought severity
even more remarkable. This is likely caused by the increasingly higher temperature under droughts, given the
exponential relationship of isoprene emissions with temperatures in MEGAN (Guenther et al., 2006).
By comparison, the modeled increase of the HCHO column with drought is 12-25%, more buffered than that of
isoprene emissions. This is mainly caused by the loss of HCHO to photolysis, which is expected to increase under
droughts with clearer skies (Wang et al., 2017; Naimark et al., 2021). In addition, HCHO formation also depends on
the abundance of oxidants, such as hydroxyl radicals (OH) and $NO_x$, that oxidize isoprene. High isoprene emissions
can suppress OH under the low-$NO_x$ conditions that prevail in part of the SE US (Wells et al., 2020), leading to the
buffered response in HCHO. Previous studies (Travis et al., 2016; Kaiser et al., 2018) showed that the NEI2011
anthropogenic inventory in the model were biased high in the SE US and a reduction of 60% of $NO_x$ emission was
suggested. By comparing to OMI $NO_2$ column, we found NoStress_GC indeed overestimates $NO_2$ columns by ~42%
in the SE US (**Figure 4b**), but the absolute bias in $NO_2$ is nearly constant from N0 to D4 (solid lines in Figure 4b).
$NO_2$ column also shows an increasing trend from N0 to D4, yet with a much smaller rate (less than 9%) than HCHO.
The model captures the relative change in $NO_2$ column with USDM (dashed lines in Figure 4b), despite the high bias
due to the NEI2011 inventory, which indicates that the changes in natural sources of $NO_x$ (e.g., biomass burning and
soil $NO_x$) with droughts are well represented by NoStress_GC. To further examine the effect of high bias of NOx on
simulated HCHO, we conducted a sensitivity simulation of reducing the NEI2011 $NO_x$ emissions by 50% over the SE
US during JJA 2011-2013. Most of the SE US was under droughts during the summertime of 2011-2012, while 2013
was a less drought-stricken year (**Figure 1**). The sensitivity simulation resulted in a small reduction of the simulated
HCHO column and the change was nearly constant among the USDM levels (**Figure S1a-b**), ranging from $-0.04 \times 10^{16}$
molec $cm^{-2}$ (2.6%) to $-0.05 \times 10^{16}$ molec $cm^{-2}$ (3.5%). This rules out the possibility that the high $NO_x$ bias in the model
is the reason for the overestimation of HCHO under droughts. Given the suppression effect of isoprene on OH and the
well-captured $NO_2$ relative changes under droughts, the overestimation of HCHO columns by the model is unlikely
to be caused by model chemistry, and more likely by the overestimation of isoprene emissions under drought
conditions.
While oxidation of anthropogenic VOCs also produces HCHO, using benzene and toluene as indicator species, we
found no change in anthropogenic VOC emissions with drought in the model (**Figure 4a**). This insensitivity rules out
anthropogenic VOCs as a key driver of model overestimation of HCHO under drought conditions. If anything, we
expect anthropogenic VOC emissions to increase during drought due to higher evaporative emissions driven by higher
temperature and more fossil fuel consumption driven by more demand for space cooling. Wildfires are another
important source that can lead to high HCHO levels, but their contributions to HCHO are more likely to be
underpredicted in GEOS-Chem partly due to insufficient hydrocarbon emissions and the underrepresented fire plume
chemistry (Alvarado et al., 2020; Liao et al., 2021; Zhao et al., 2022). A deeper planetary boundary layer (PBL) is
expected under droughts primarily caused by a larger sensible height flux released from dry soil (Miralles et al., 2014).
Indeed, the MERRA-2 PBL height used in our simulation increases by 12.42%, 17.79%, 20.99%, 26.21%, and 29.52%
from D0 to D4 relative to the value of 1589 m at N0 in the SE US during the midday (13:30 LT). Considering the
PBL heights in MERRA-2 agree well with observations with only an overall 200 m low bias (Guo et al., 2021), we
do not expect mixing heights to be the main cause of the high bias of HCHO column under drought conditions. To
further quantify the effects of wildfires and PBL on the changes of HCHO column with drought, we conducted two
additional sensitivity tests: (1) turning off the GFED4 wildfire emission inventory during 2011-2013 JJA, and (2)
keeping PBL constant as in 2013 (normal year) during 2011-2012 (drought years) JJA. The results in **Figure S1c-d**
show overall negligible changes in HCHO column in the SE US, which verifies our assumptions above.
In summary, the model overestimates HCHO increases during drought as compared to OMI. This overestimation is
attributed to the model overestimation of isoprene emissions during drought. Drought stress effect on isoprene
emissions is thus required in GEOS-Chem to resolve the discrepancy in HCHO responses to drought between OMI
and the model.

**3.2 Isoprene flux measurement**

To further evaluate isoprene emissions in NoStress_GC, we compared the isoprene flux measurements at the
MOFLUX site (Potosnak et al., 2014; Seco et al., 2015) with predicted isoprene emissions at the model grid that
contains the site. At the time of writing, the MOFLUX site is the only long-term, canopy-level, biogenic isoprene flux
measurement site in the Northern midlatitude that sampled droughts. The site experienced multiple drought levels in
the summer of 2012, which allows for the model-observation comparison across different drought severities as shown
in **Figure 5**. The abnormal dry conditions (D0) started in early June, which developed to moderate drought (D1) in
late June, worsened to severe drought (D2) and extreme drought (D3) in July-August, and bounced back to D2 in
September (**Figure 5a**). The model generally captures the daily variability of isoprene emissions with a statistically
significant correlation coefficient (R) of 0.67, but its biases differ by USDM levels. The model underestimates isoprene
flux from N0 (bias of -1.81 mg/m$^2$/hr) to D1 (bias of -2.89 mg/m$^2$/hr), has a minimal bias (-0.47 mg/m$^2$/hr) at D2, and
changes to an overestimate at D3 (bias of 1.2 mg/m$^2$/hr) (**Figure 5b**). While differences are expected when comparing
a single-point flux measurement with the grid-mean model prediction, such differences most likely result in a
systematic bias that should not relate to the temporal variability of drought. The fact that the model bias changes from
being underpredicting to overpredicting as drought severity increases further confirms the importance of the model
lack of a drought suppression effect on isoprene emissions during severe to exceptional droughts (D3 and D4). This
is qualitatively consistent with that of the HCHO biases described above.

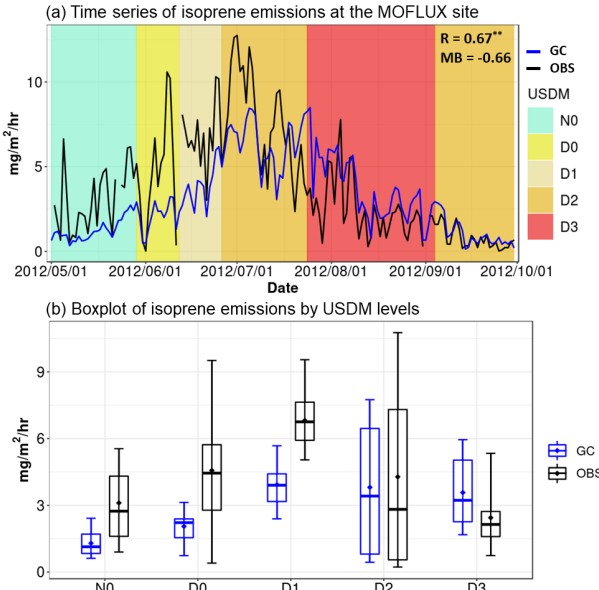

**Figure 5. (a) Comparison of daily time series of isoprene emissions observed at the MOFLUX site (OBS) and simulated by MEGAN2.1 in GEOS-Chem (GC). The background is color-coded according to the USDM drought severity. R and MB at the upright corner show the correlation coefficient and mean bias, respectively. (b) Boxplot of isoprene emissions separated by USDM drought levels. The upper and lower whiskers represent the 90% and 10% quantiles, respectively.**

## 4. Drought Stress Algorithm

The MEGAN2.1 isoprene emission routines in GEOS-Chem use a simplified mechanistic representation of the major environmental factors controlling biogenic emissions (Guenther et al., 2012), in which the isoprene emission factor $\gamma_{2.1}$ is the product of a canopy-related normalization factor ($C_{FAC}$) multiplied by other factors representing light ($\gamma_{PAR}$), temperature ($\gamma_T$), leaf age ($\gamma_{AGE}$), LAI ($\gamma_{LAI}$), carbon dioxide ($CO_2$) inhibition ($\gamma_{CO2}$), and soil moisture ($\gamma_{SM}$):

$$\gamma_{2.1} = C_{FAC}\gamma_{PAR}\gamma_T\gamma_{AGE}\gamma_{LAI}\gamma_{CO2}\gamma_{SM} = \gamma_0\gamma_{SM} \tag{1}$$

where $\gamma_0$ is the product of the non-drought factors. Because of the lack of reliable soil moisture databases, $\gamma_{SM}$ is always set to be one in GEOS-Chem as in many other chemical transport models, which means no water stress term in the standard model configuration (i.e., NoStress_GC). We show above that NoStress_GC overestimates isoprene emissions and consequently HCHO column densities under drought conditions in the SE US. In this section, we describe the approach whereby observational constraints from the MOFLUX isoprene flux measurement and OMI HCHO were separately used to derive a drought stress factor $\gamma_d$ which replaces $\gamma_{SM}$ in Equation (1) to simulate the response of isoprene emissions to drought in MEGAN2.1 implementation of GEOS-Chem (hereafter referring to as GC/MEGAN2.1). The drought stress factor $\gamma_d$ derived from the MOFLUX isoprene flux measurement is denoted as $\gamma_{d\_MOFLUX}$ and that from OMI HCHO as $\gamma_{d\_OMI}$. Their corresponding simulations are referred to as MOFLUX_Stress_GC and OMI_Stress_GC, respectively. In either algorithm, the underlying assumption is that the GEOS-Chem model has no significant bias in predicting isoprene fluxes or HCHO columns due to factors other than isoprene emissions under drought conditions. The assumption is reasonable because the GEOS-Chem model uses

reanalysis meteorology, state-of-the-science isoprene oxidation schemes, time-specific anthropogenic emissions and
fire emissions, and natural emissions calculated online using model meteorology as described in Section 2.3. The
discussion in Section 3.1 validated some aspects of the assumption such as NOx emissions, wildfire emissions, and
PBL.

**4.1 MOFLUX-based Drought Stress Algorithm**

The $\gamma_{d\_MOFLUX}$ was derived following Jiang et al. (2018) by implementing photosynthesis and water stress parameters
with a formula of:
$$\gamma_{d\_MOFLUX} = \gamma_0 \gamma_{d\_isoprene} \begin{cases} \gamma_{d\_isoprene} = 1 \ (\beta_t \geq 0.3) \\ \gamma_{d\_isoprene} = V_{cmax}/\alpha \ (\beta_t < 0.3, \alpha = 77) \end{cases} \tag{2}$$
where $V_{cmax}$ is the maximum carboxylation rate by photosynthetic Rubisco enzyme and $\beta_t$ represents the water stress
ranging from zero (fully stressed) to one (no stress). This simplified method intends to use the decreased
photosynthetic enzyme activity to physiologically represent the variation in isoprene emissions under drought stress
via dividing $V_{cmax}$ by an empirical parameter $\alpha$ when the water stress is below a threshold.
Since the default GEOS-Chem does not have these photosynthetic parameters, we adopted the ecophysiology module
created by Lam et al. (2022) that is based on the photosynthesis calculation in the Joint UK Land Environmental
Simulator (JULES; Best et al., 2011; Clark et al., 2011) as an online component in GEOS-Chem so that it simulates
photosynthesis rate and bulk stomatal conductance dynamically and consistently with the underlying meteorology that
drives GEOS-Chem. The outputs of $V_{cmax}$ and $\beta_t$ from the ecophysiology module were passed to MEGAN2.1 in GEOS-
Chem to parameterize the drought stress according to Equation 2. In addition to GEOS-Chem meteorology, the
ecophysiology module uses soil parameters from the Hadley Centre Global Environment Model version 2 – Earth
System Model (HadGEM2-ES). In general, the implementation of the ecophysiology module much improved the
simulated stomatal conductance and dry deposition velocity relative to site observations on average for seasonal
timescales, but the $\beta_t$ computed has not been calibrated to intermittent drought conditions. Instead of adopting the
values of $V_{cmax}$ and $\beta_t$ from Jiang et al. (2018) which were based on the Community Land Model, we need to determine
the $\beta_t$ threshold and the $\alpha$ value specific to GEOS-Chem with the ecophysiology module of Lam et al. (2022). To
calibrate $\beta_t$, we first examined the statistical distribution of $\beta_t$ at the MOFLUX grid (**Figure S2**) during May-September
2011 and 2012 when multiple USDM drought categories occurred. Then we decided on a value of 0.3 as the threshold
$\beta_t$ below which the drought stress will be triggered in the model because this value is greater than 75% quantile of all
the $\beta_t$ values from D0 to D3, thus capturing most of the drought conditions.
We note the observed isoprene flux at MOFLUX is consistently higher than predicted values during the non-drought
period (e.g., N0 in Figure 5a). This systematic bias is expected because we compare the single-point observations with
grid-mean isoprene emission fluxes. To correct the systemic bias, we scaled down the model isoprene emissions at
the MOFLUX grid by a factor of 1.42, which is the ratio of the average hourly isoprene fluxes between observations
and simulations at the MOFLUX grid during non-drought conditions ($\beta_t > 0.3$). The factor of 1.42 was applied to
downscale modeled isoprene fluxes at the MOFLUX grid during the entire time series, including drought conditions.
The resulted time series are shown in **Figure 6a**. Based on the downscaled model prediction, we derived that α=77
under drought conditions ($\beta_t$ < 0.3), which minimized the mean bias under drought conditions between the modeled
and observed isoprene fluxes at the MOFLUX grid.
**Figure 6b** shows the comparison of the hourly NoStress_GC and MOFLUX_Stress_GC isoprene emissions with
observations in May-September 2012. The overall mean bias is reduced from 2.05 mg/m²/hr to 0.01 mg/m²/hr despite
the fact that the stress factor is only applied to drought conditions. The correlation coefficient (R) and index of
agreement (IOA) also increase from 0.77 to 0.85 and from 0.80 to 0.93, respectively. All the changes in the comparison
metrics indicate the model simulations are improved considerably based on the single-point measurement.

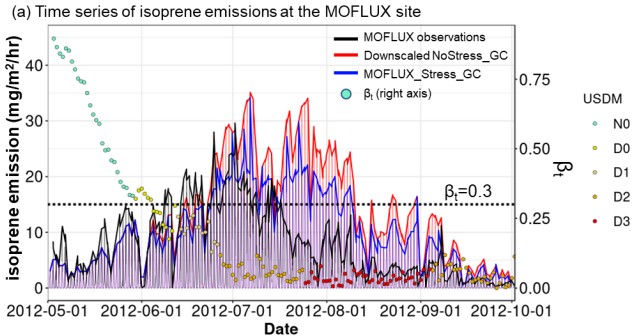

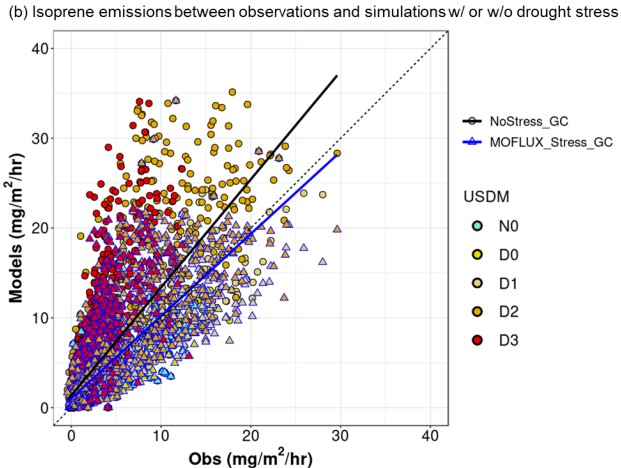


**Figure 6. (a) Hourly time series of isoprene emissions at the MOFLUX site from observations (black line) and simulations**
**with (MOFLUX_Stress_GC; blue line) and without drought stress (NoStress_GC; red line; after downscaling). The dots**
**color-coded by USDM levels represent the daily values of $\beta_t$ (right axis). The dashed line indicates the threshold of 0.3. (b)**
**Comparison of isoprene emissions between observations (Obs) and simulations with (MOFLUX_Stress_GC; blue-bordered**
**triangle) and without (NoStress_GC; black-bordered circle) drought stress. Data are color-coded by USDM levels.**

 **4.2 OMI-based Drought Stress Algorithm**

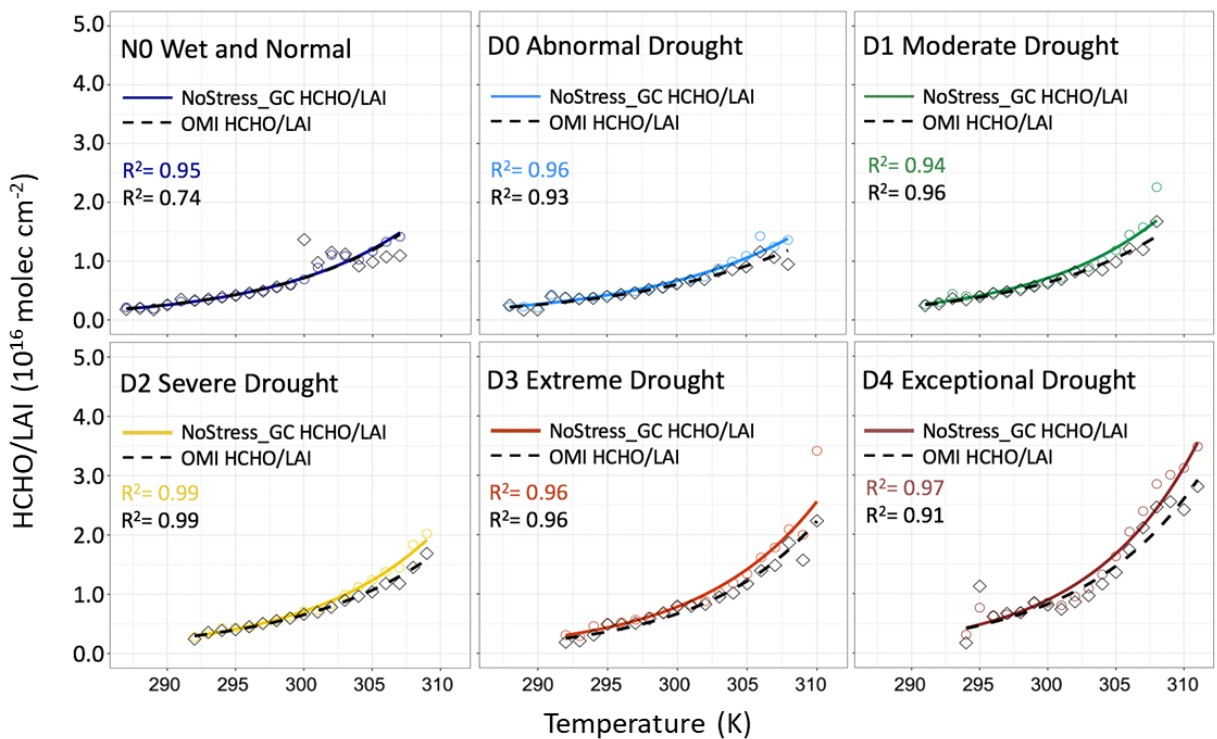


**Figure 7. Response of HCHO/LAI ratio ($10^{16}$ molec cm$^{-2}$) to temperature (K) in different drought levels averaged over JJA 2005-2017. The colored solid line is the modelled NoStress_GC HCHO/LAI ratio, and the black dashed line is the observed HCHO/LAI ratio from OMI. The exponentially fitted formulas and the resulted coefficient of determination ($R^2$) are labelled in each subplot.**

Isoprene emission increases exponentially with temperatures below ~310 K (Guenther et al., 2006) in the absence of other stress factors such as drought. Indeed, an exponential relationship between biogenic isoprene emission per unit LAI and temperature is predicted by MEGAN2.1 at all USDM levels (**Figure S3**). However, the predicted temperature sensitivity is found to increase substantially with drought severity with no sign of plateauing or slow-down even under the most severe drought conditions when MOFLUX measurements measured a decrease in isoprene emissions (c.f. Figure 5). Similarly, we found NoStress_GC overestimates HCHO sensitivities to high temperatures (> 300 K) under drought conditions (D0-D4) (**Figure 7**), but no such overestimation is seen under non-drought (N0) or low temperature conditions during drought (< 300 K). This indicates the role of drought stress on isoprene emissions is likely through suppressing the dependence of emissions on temperatures during drought. Leaf level measurements conducted during the 2012 drought at the MOFLUX site provide independent evidence of drought suppression of the isoprene response to increasing temperature for less drought-resilient tree species (Geron et al., 2016). Taking advantage of these empirical observations, we derived the OMI-based drought stress algorithm by minimizing the differences in HCHO column sensitivities to temperatures between OMI and GEOS-Chem under drought conditions as shown in **Figure 7**. When calculating the relationships between HCHO column densities and temperatures, we first scaled HCHO column by LAI on a grid-by-grid basis to account for the regional differences in isoprene emissions due to different vegetation coverage as well as the effect of LAI changes with drought (c.f. **Figure 4**). Each point in **Figure 7** represents the mean

HCHO/LAI ratio, denoted as $\Omega$, within each 1K temperature interval. We used exponential functions ($ln\Omega = kT + b$) to
separately fit the temperature (T) dependence of HCHO/LAI ratio ($\Omega$) under different drought levels (**Figure 7**) for
both the model and OMI. The resulting formulas were listed in **Table 1** and the $R^2$ of most fitting lines is greater than

386 0.9.

**Table 1. Fitted exponential formulas of NoStress_GC and OMI HCHO/LAI ratio ($\Omega$, $10^{16}$ molec cm$^{-2}$) to surface air**
**temperature (T, K), and fitted value of HCHO/LAI ratio at 290K, 300K, and 310K.**

| USDM | NoStress_GC HCHO/LAI ($\Omega$, $10^{16}$ molec cm$^{-2}$) | | | | OMI HCHO/LAI ($\Omega$, $10^{16}$ molec cm$^{-2}$) | | | |
|------|-----------------|------|------|------|-----------------|------|------|------|
| | Fitting Formula | 290K | 300K | 310K | Fitting Formula | 290K | 300K | 310K |
| N0 | $ln\Omega = 0.104T - 31.42$ | 0.25 | 0.72 | 2.03* | $ln\Omega = 0.101T - 30.78$ | 0.26 | 0.72 | 1.97* |
| D0 | $ln\Omega = 0.091T - 27.83$ | 0.27 | 0.67 | 1.66 | $ln\Omega = 0.085T - 25.92$ | 0.26 | 0.60 | 1.40 |
| D1 | $ln\Omega = 0.108T - 32.83$ | 0.24 | 0.71 | 2.10 | $ln\Omega = 0.100T - 30.56$ | 0.23 | 0.64 | 1.74 |
| D2 | $ln\Omega = 0.110T - 33.33$ | 0.24 | 0.71 | 2.14 | $ln\Omega = 0.098T - 29.97$ | 0.24 | 0.65 | 1.75 |
| D3 | $ln\Omega = 0.118T - 35.72$ | 0.24 | 0.78 | 2.56 | $ln\Omega = 0.121T - 36.62$ | 0.20 | 0.67 | 2.23 |
| D4 | $ln\Omega = 0.125T - 37.59$ | 0.26 | 0.90 | 3.13 | $ln\Omega = 0.115T - 34.62$ | 0.26 | 0.83 | 2.60 |

* Asterisk indicates that the temperature does not reach this value in actual data and is an extrapolated value.
As the fitting equations suggest, both NoStress_GC and OMI HCHO/LAI ratios increase with temperature under all
conditions, but the former shows a higher sensitivity to temperature under drought conditions. This can be clearly seen
from the higher HCHO/LAI ratios of NoStress_GC ($\Omega_{GC}$; solid lines) than those of OMI ($\Omega_{OMI}$; dashed lines)
especially when the temperature is greater than 300 K under D0-D4. To better explain this, we also calculated the
fitted value of HCHO/LAI at three temperatures of 290K, 300K, and 310K in **Table 1**. Since it is difficult for the N0
condition to reach a temperature of 310K, the values were extrapolated and marked with an asterisk in the table. The
results show that the model overestimates the temperature dependence at all drought levels. At 290K, all biases
between $\Omega_{OMI}$ and $\Omega_{GC}$ are less than $0.05 \times 10^{16}$ molec cm$^{-2}$. At 310K, the bias between the two is $0.06 \times 10^{16}$ molec
cm$^{-2}$ (3.0%) at N0 but increases by more than a factor of 4 to $0.26 \times 10^{16}$ molec cm$^{-2}$ (18.6%), $0.36 \times 10^{16}$ molec cm$^{-2}$
(20.7%), $0.39 \times 10^{16}$ molec cm$^{-2}$ (22.3%), $0.33 \times 10^{16}$ molec cm$^{-2}$ (14.8%), and $0.53 \times 10^{16}$ molec cm$^{-2}$ (20.4%) at D0-
D4 drought, respectively. As isoprene emission is a fixed function of temperature in MEGAN2.1, the overdependence
of HCHO column on temperature is caused by the previous two weeks' temperatures being higher under drought,
which leads to a higher value of $\gamma_T$ reflecting the temperature "memory" effects on isoprene emissions (**Figure S4**).
Based on the fitted formulas in **Table 1**, the ratio between $\frac{\Omega_{OMI}}{\Omega_{GC}}$ under each level from D0 to D4 can be derived by:
$$\frac{\Omega_{OMI}}{\Omega_{GC}} = \frac{e^{k_{OMI}T + b_{OMI}}}{e^{k_{GC}T + b_{GC}}} = e^{(k_{OMI} - k_{GC})T} e^{(b_{OMI} - b_{GC})} \qquad (3)$$
where $k_{OMI}$ ($k_{GC}$) and $b_{OMI}$ ($b_{GC}$) represent the slopes and interpolations of the formulas in **Table 1** for OMI (GC)
HCHO column; T is surface temperature, and e is the exponential constant. By averaging the values of $k_{OMI}$-$k_{GC}$ and
$b_{OMI}$-$b_{GC}$ from D0 to D4, we can obtain:
$$\frac{\Omega_{OMI}}{\Omega_{GC}} = 380.10 e^{-0.02T} \ (\beta_t < 0.6, T > 300 \ K) \qquad (4)$$
where $\beta_t < 0.6$ represents the 75% quantile of the $\beta_t$ values from D0 to D4 for the whole SE US study region in JJA
2005-2017 (**Figure S2**).
The formula of $\gamma_{d\_OMI}$ is thus:
$$\gamma_{d\_OMI} = \gamma_0 \gamma_{d\_isoprene} \begin{cases} \gamma_{d\_isoprene} = 1 \ (\beta_t \geq 0.6 \ or \ T \leq 300K) \\ \gamma_{d\_isoprene} = \frac{\Omega_{OMI}}{\Omega_{GC}} = 380.10 e^{-0.02T} \ (\beta_t < 0.6, T > 300K) \end{cases} \qquad (5)$$
Note the threshold of $\beta_t$ in equation 5 is different from the value used by $\gamma_{d\_MOFLUX}$ because all the SE US grids were
considered in deriving $\beta_t$ for $\gamma_{d\_OMI}$. Another difference is that the factor is activated only if the temperature is higher
than 300K when significant biases between $\Omega_{OMI}$ and $\Omega_{GC}$ are found (**Figure 7**).

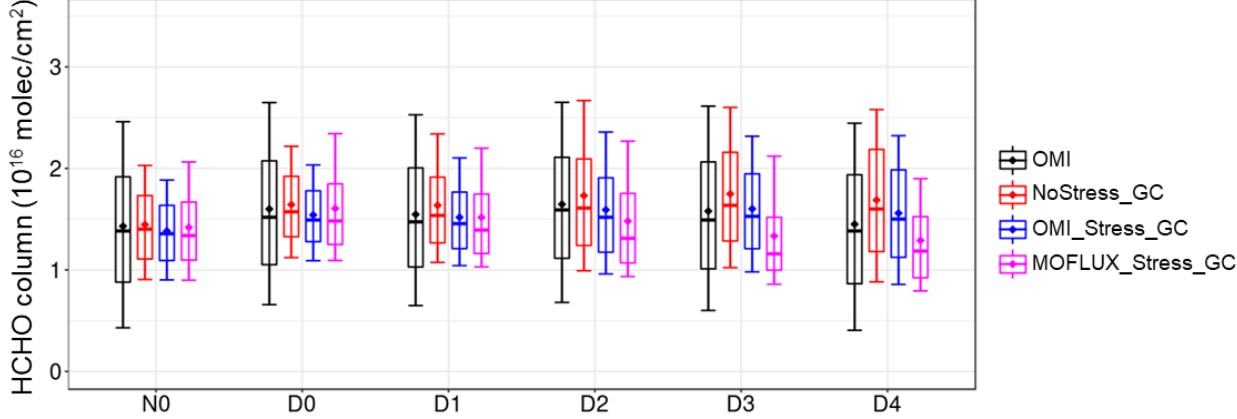


**Figure 8. Boxplot of HCHO column statistical distributions for OMI observations (black) and different GEOS-Chem**
**simulations: without drought stress (NoStress_GC; red) and with drought stress factors derived from MOFLUX**
**observations (MOFLUX_Stress_GC; blue) and from OMI HCHO constraints (OMI_Stress_GC; pink).**
**Figure 8** compares the statistical distributions of HCHO column densities from OMI, NoStress_GC,
MOFLUX_Stress_GC, and OMI_Stress_GC during May-September 2012 over the SE US. Compared to OMI,
NoStress_GC simulation has a mean high bias of $0.02 \times 10^{16}$ molec cm$^{-2}$ - $0.24 \times 10^{16}$ molec cm$^{-2}$ during D0-D4. The
$\gamma_{d\_OMI}$ algorithm reduces the high bias to $-0.05 \times 10^{16}$ molec cm$^{-2}$ - $0.11 \times 10^{16}$ molec cm$^{-2}$. By contrast, the $\gamma_{d\_MOFLUX}$
algorithm reduces the HCHO simulations too much over the SE US and causes an overall underestimation of $0.02 \times 10^{16}$
molec cm$^{-2}$ - 0.25×10$^{16}$ molec cm$^{-2}$. The $\gamma_{d\_MOFLUX}$ algorithm also narrows the statistical distribution of HCHO as
indicated by the smaller boxes and shorter whiskers compared to OMI. This suggests that the $\gamma_{d\_MOFLUX}$ algorithm
based on the single-site observations is incapable of representing the drought stress over the SE US, possibly because
the MOFLUX site has thin soil layers and thus is vulnerable to water stress (Opacka et al., 2022). Isoprene emissions
measured here are therefore more sensitive to droughts and the same extent of drought stress is likely too strong to be
applied to other regions in the SE US. As a result, the $\gamma_{d\_OMI}$ algorithm is used in the next section to further evaluate
how this algorithm would change the responses of atmospheric compositions to droughts.
**5. Changes in Simulated Biogenic Isoprene Emissions, HCHO, O$_3$, and OA**
In this section, we evaluated the changes in biogenic isoprene emissions and HCHO column densities by running a
long-term (2005-2017, JJA) simulation, after adding the OMI-based drought stress factor for isoprene emissions $\gamma_{d\_OMI}$
in GEOS-Chem. Since isoprene is an important precursor for the formation of tropospheric O$_3$ and OA, maximum
daily 8-hour average (MDA8) O$_3$, and OA changes were also examined. We used the ComplexSOA mechanism in
GEOS-Chem (Pye et al., 2010; Marais et al., 2016) which includes more detailed pathways of isoprene to secondary
organic aerosols such as aqueous-phase reactive uptake and the formation of organo-nitrates.
**Figure 9** shows the changes in biogenic isoprene emissions resulting from adding $\gamma_{d\_OMI}$ drought stress in GEOS-
Chem. Here we expanded the maps to the entire contiguous US to examine whether the drought stress algorithm can
impose large changes on other US regions although such changes need to be interpreted with caution. The numbers at
each panel indicate the means of isoprene emissions of NoStress_GC and the mean differences (MD) relative to the
OMI_Stress_GC over the SE US. As expected, the biggest decrease in isoprene emissions is found in the SE US with
the regional-mean emissions reduced by 0.17×10$^{-10}$ kg m$^{-2}$ s$^{-1}$ (8.60%), 0.35×10$^{-10}$ kg m$^{-2}$ s$^{-1}$ (14.24%), 0.43×10$^{-10}$ kg
m$^{-2}$ s$^{-1}$ (16.57%), 0.49×10$^{-10}$ kg m$^{-2}$ s$^{-1}$ (17.49%), 0.58×10$^{-10}$ kg m$^{-2}$ s$^{-1}$ (18.66%), and 0.65×10$^{-10}$ kg m$^{-2}$ s$^{-1}$ (20.74%)
from N0 to D4, respectively (**Figure 9c**). Despite lowering emissions relative to NoStress_GC, OMI_Stress_GC
simulates an increase of isoprene emissions under drought conditions compared to non-drought in the SE US; the
respective increases are 0.28×10$^{-10}$ kg m$^{-2}$ s$^{-1}$ (15.20%), 0.34×10$^{-10}$ kg m$^{-2}$ s$^{-1}$ (18.40%), 0.49×10$^{-10}$ kg m$^{-2}$ s$^{-1}$ (26.47%),
0.69×10$^{-10}$ kg m$^{-2}$ s$^{-1}$ (37.46%), and 0.65×10$^{-10}$ kg m$^{-2}$ s$^{-1}$ (35.23%) from D0 to D4 relative to N0 (**Figure 9c**). This
increase results from the top-down constraints by the corresponding changes in OMI HCHO column densities with
USDM and consequently exhibits the behavior of non-uniform increases with drought severity (e.g., peak increase of
37.5% at D3, followed by a ~2% reduction at D4), which is consistent with the MOFLUX flux measurements.
For other regions, such as California and Minnesota, biogenic isoprene emissions decreased slightly by less than
0.5×10$^{10}$ kg m$^{-2}$ s$^{-1}$. The smaller effect of the drought stress factor imposed on regions other than the SE US is
understandable because of the lower isoprene emissions.

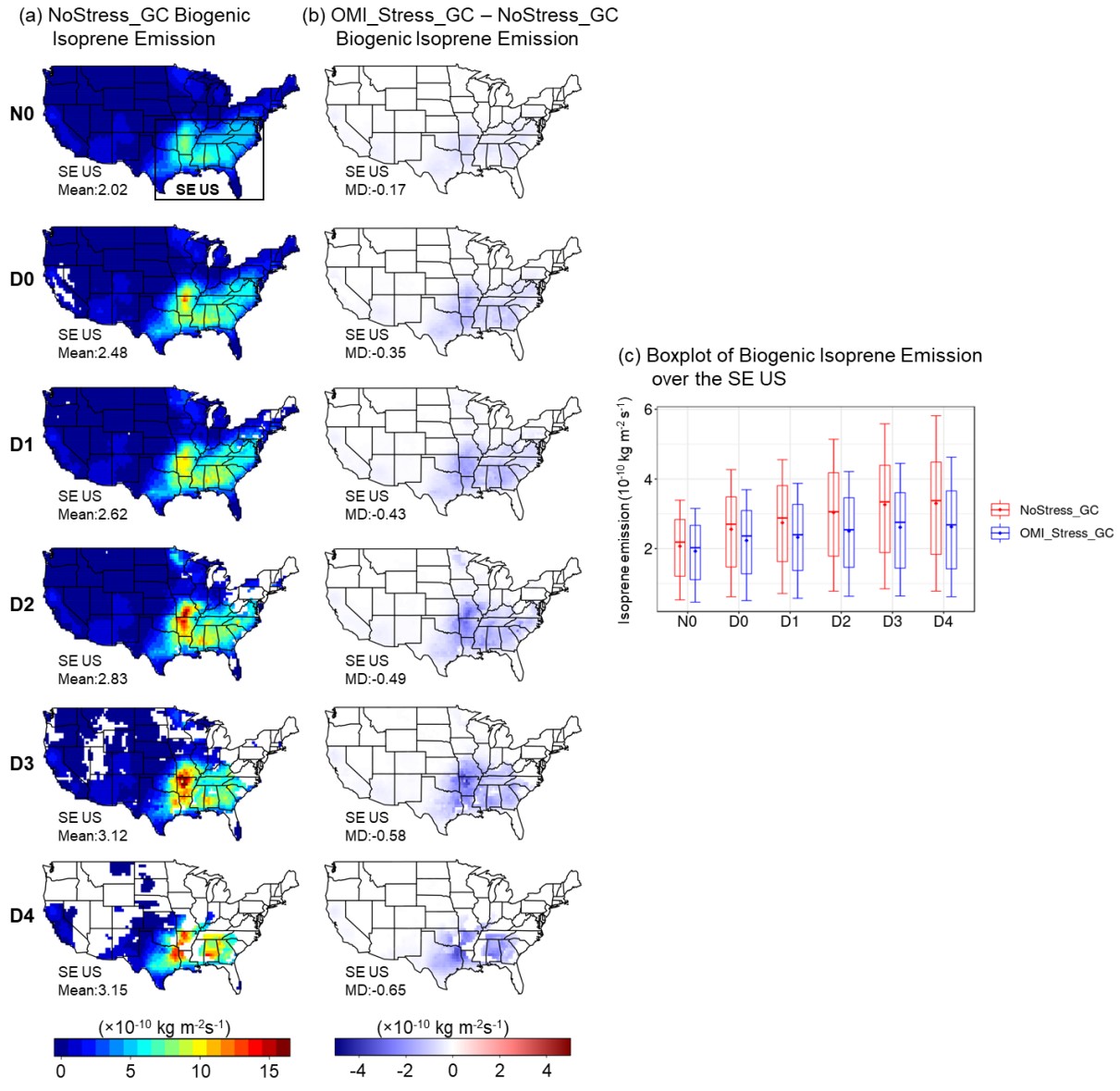


**Figure 9. Simulated biogenic isoprene emissions during JJA 2005-2017 by USDM dryness category by NoStress_GC (a),**
**OMI_Stress_GC minus NoStress_GC (b), and statistical distributions of SE US isoprene emissions between the two**
**simulations (c). Numbers at the bottom-left corner of each panel indicate the SE US (black box) regional mean of biogenic**
**isoprene emissions for NoStress_GC (left column), and mean differences (MD) between OMI_Stress_GC and NoStress_GC**
**(middle column).**

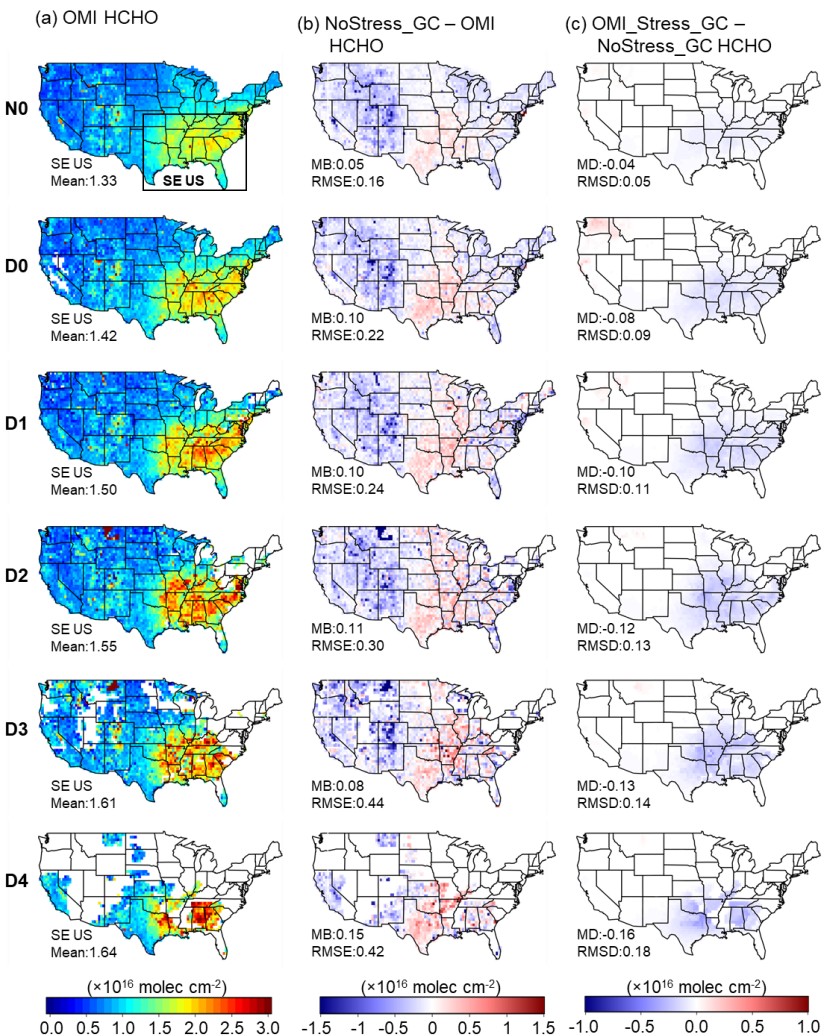

462

**Figure 10. Mean HCHO column densities during JJA 2005-2017 by USDM dryness category for OMI (a), NoStress_GC minus OMI (b), and OMI_Stress_GC minus NoStress_GC (c). Numbers at the bottom-left corner of each panel indicate the SE US (black box) regional mean of OMI HCHO column (left column), mean bias (MB), and root mean square error (RMSE) in HCHO column densities between NoStress_GC and OMI (middle column), and mean differences (MD) and root mean square deviation (RMSD) between OMI_Stress_GC and NoStress_GC (right column). MD and RMSD are calculated in the same way as MB and RMSE; the different names are used to distinguish between model-to-model comparison and model-to-observation comparison, respectively.**

The changes in the HCHO column are shown in **Figure 10**. Different from the overestimation in the SE US, NoStress_GC underestimates HCHO column densities in the western US compared to OMI (**Figure 10b**). This negative bias should be interpreted with care because the scaling factor of 1.5 (c.f. section 2.2) is derived over the SE US and may not hold in other regions. For the SE US overall, the drought stress factor reduces modeled HCHO columns by $0.08 \times 10^{16}$ molec cm$^{-2}$ (5.43%), $0.10 \times 10^{16}$ molec cm$^{-2}$ (6.46%), $0.12 \times 10^{16}$ molec cm$^{-2}$ (7.22%) and $0.13 \times 10^{16}$ molec cm$^{-2}$ (7.62%), $0.16 \times 10^{16}$ molec cm$^{-2}$ (8.91%) under D0-D4, respectively, relative to NoStress_GC (**Figure 10c**). This leads to a better agreement with OMI as OMI_Stress_GC has nearly zero MB under D0-D4 (**Figure S5**; MB = $-0.05 \times 10^{16}$ molec cm$^{-2}$ $\sim 0.02 \times 10^{16}$ molec cm$^{-2}$). The RMSE is also reduced by 3%-13% relative to the NoStress_GC simulation compared to observations. The changes in both metrics indicate that the drought algorithm

considerably improves the model performance in capturing the biogenic isoprene response to drought as evidenced by
HCHO column. Similar to the changes in biogenic isoprene emissions, the OMI_Stress_GC only slightly decreases
HCHO column densities (<5%) compared to the NoStress_GC simulation in other US regions.

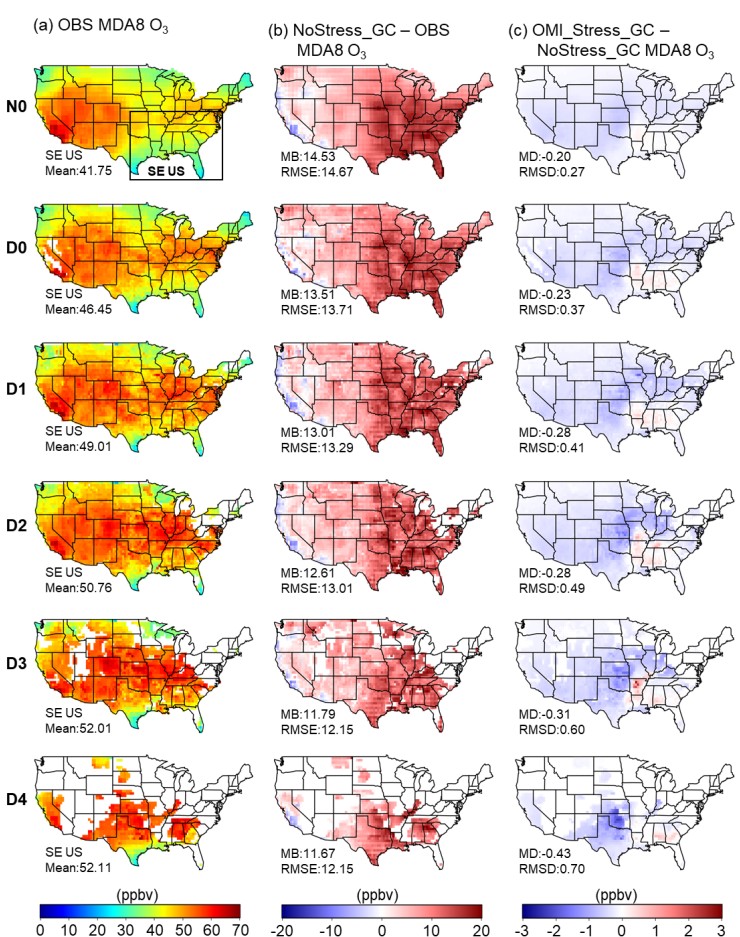


**Figure 11. Same as Figure 10 but for surface maximum daily 8-hour average (MDA8) O₃.**
**Figure 11a** displays the observed MDA8 $O_3$ changes with USDM. Similar to the changes of the HCHO column with
USDM levels, $O_3$ in the SE US exhibits a gradual increase, relative to the mean of 41.74 ppbv at N0, of 4.70 ppbv,
7.26 ppbv, 9.01 ppbv, 10.26 ppb, and 10.36 ppbv under D0-D4, respectively. This is consistent with our previous
study (Li et al., 2022; Lei et al., 2022) which investigated $O_3$ changes with drought severity in more detail. The
NoStress_GC simulation has a high bias in MDA8 $O_3$ across all USDM categories (**Figure 11b**). High positive bias
is a common issue of surface $O_3$ simulations in chemical transport models, which is a research question and can be
attributed to the uncertainties in various processes, such as $NO_x$ emissions, isoprene oxidation pathways, $O_3$ dry
deposition velocity, boundary layer dynamics (Fiore et al., 2005; Lin et al., 2008; Squire et al., 2015; Travis et al.,
2016; Travis and Jacob, 2019). Despite the systematic high bias, NoStress_GC captures the increasing trend of MDA8
$O_3$ with increasing dryness but with a respectively smaller increment (relative to N0) of 3.62 ppbv, 5.67 ppbv, 7.01
ppbv, 7.41 ppbv, and 7.41 ppbv under D0 to D4. This discrepancy between NoStress_GC and observations can also

be inferred from the fact that the MB between model and observations decreases from 14.53 ppbv at N0 to 11.67 ppbv at D4 (**Figure 11b**). **Figure 11c** shows the difference in MDA8 $O_3$ between OMI_Stress_GC and NoStress_GC. In the SE US where isoprene emissions are the highest and reduced the most by the drought stress algorithm, OMI_Stress_GC shows a small increase in MDA8 $O_3$ of less than 1 ppbv. This increase in $O_3$ can be explained by an increase of OH resulting from reducing isoprene emissions under low-$NO_x$ conditions in the SE US (Wells et al., 2020). For the SE US study domain as a whole, the change in MDA8 ozone was negligible but negative (regional mean of -0.5 ppbv). Although the drought factor does not reduce the overall high bias, it makes the model more consistent with the observed increment in MDA8 $O_3$ for the subregion with increased $O_3$ (e.g., 90–94°W, 32–35°N) as drought severity increases. Since $NO_x$ has a high positive bias from the NEI2011 inventory (**Figure 4**), the improvement of MDA8 in these regions is likely to be underestimated. Over northeastern Texas, Oklahoma, and Kansas where isoprene emission is also reduced by the drought algorithm yet from a much lower emission base compared to other SE US areas, OMI_Stress_GC simulates 1-3 ppbv lower MDA8 $O_3$ under drought conditions (D0-D4), leading to a better agreement with observations. For regions with lower isoprene and higher $NO_x$ concentrations, $O_3$ formation is more sensitive to the changes in isoprene, which explains the reduction in MDA8 $O_3$ caused by the drought stress factor.

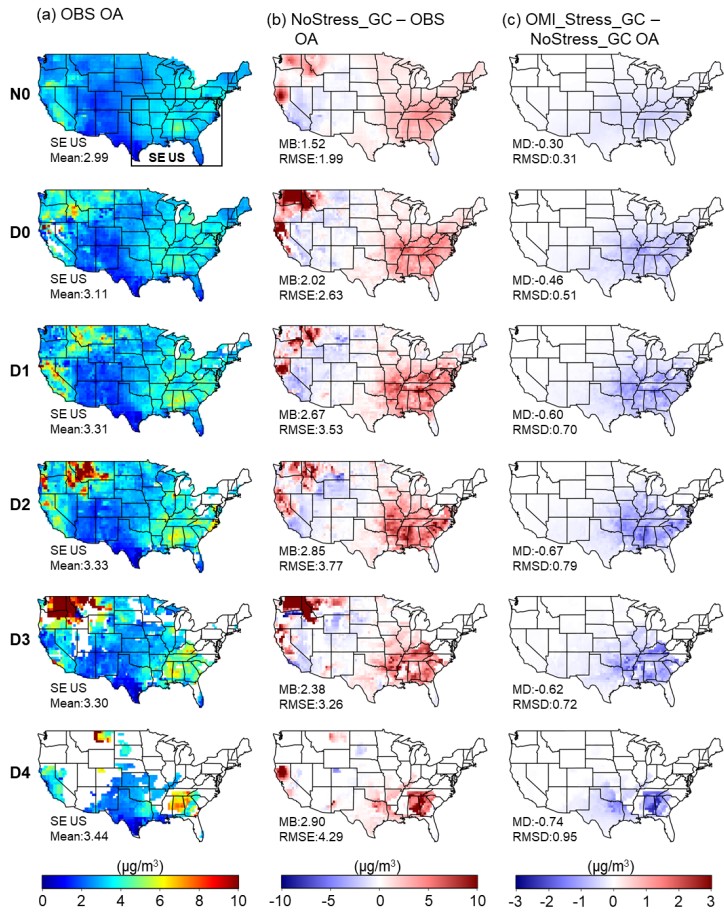

**Figure 12. Same as Figure 10 but for organic aerosol (OA).**

The changes in OA with USDM are shown in **Figure 12.** Observed OA in the SE US shows an average increase
(relative to N0) of 0.12 µg/m$^3$, 0.32 µg/m$^3$, 0.34 µg/m$^3$, 0.31 µg/m$^3$, and 0.45 µg/m$^3$ under D0 to D4, respectively. The
extremely high values over the northwest states (e.g., Washington and Montana) are likely associated with higher
wildfire emissions under droughts (Wang et al., 2017). The NoStress_GC simulation considerably overestimates OA
in the SE US with an MB of 1.52 µg/m$^3$ (50.83%) at N0 and the overestimation becomes even higher to 2.02-2.90
µg/m$^3$ (64.95%-85.58%) at D0-D4 (**Figure 12b**), thus causing an overprediction of the drought-OA relationship.
Zheng et al (2020) reported a similar level of overestimation and attributed this to the overdependence of isoprene-
derived secondary organic aerosol (SOA) on sulfate. As isoprene is one of the dominant sources of OA in the SE US
(Xu et al., 2015; Budisulistiorini et al., 2016), our analysis suggests that the model overestimation of isoprene
emissions under drought conditions is another reason for this high OA bias in the SE US. Indeed, the drought stress
factor greatly improves the OA simulation by reducing the MB by 0.30 µg/m$^3$ (6.60%), 0.46 µg/m$^3$ (8.98%) 0.60
µg/m$^3$ (10.07%), 0.67 µg/m$^3$ (10.85%), 0.62 µg/m$^3$ (10.88%), 0.74 µg/m$^3$ (11.71%) under N0 to D4 over the SE US
relative to NoStress_GC, thus lowering the MB to be within 1.22-2.18 µg/m$^3$ (40.82% - 65.52%; **Figure S5**) compared
with observations. We also examined the change of three major SOA components in **Figure S6**. Anthropogenic SOA
(ASOA) barely changes; isoprene SOA (ISOA) decreases the most as expected since the drought stress factor is
applied to isoprene emissions only. Interestingly, terpene SOA (TSOA) also shows a slight decrease, suggesting
positive feedback between ISOA and TSOA.
In summary, the OMI-based drought stress factor shows good performance in correcting the overestimation of
biogenic isoprene in default GEOS-Chem simulations under drought conditions. The drought stress factor was
constrained by the observed exponential fitting between the HCHO to LAI ratio and temperature, not by observed
HCHO columns directly. It nearly eliminates the high HCHO bias compared with OMI observations in the SE US
under drought conditions, which consequently improves the simulation of OA. MDA8 O$_3$ slightly increases in the
areas with high isoprene emissions, leading to no improvement in model bias but a better agreement with the observed
O$_3$ increment with drought severity. Places with lower isoprene emissions show an MDA8 O$_3$ reduction of 1-3 ppbv,
indicating the region-specific O$_3$ responses to the changes of isoprene due to the nonlinearity of O$_3$ chemistry.
**6. Conclusions**
Using long-term (JJA 2005-2017) weekly USDM drought index and OMI HCHO column data over the SE US, we
revealed a step-increase pattern of HCHO by 6.7%, 12.6%, 16.5%, 21.2%, and 23.2% from D0 to D4 relative to non-
drought conditions (N0), respectively, which indicates the increasingly higher isoprene emissions with drought on a
regional scale although the rate of increase decreases under severe droughts. Compared with OMI observations, the
GEOS-Chem simulated HCHO column density exhibits a similar pattern, but the changes are 1.1-1.5 times higher
with a respective increase of 9.90%, 15.1%, 19.5%, 21.8%, and 29.1% from D0 to D4. Since there are no big changes
in anthropogenic VOCs under droughts, biogenic isoprene emissions are the key drivers for the increase of HCHO,
and a drought stress factor is missing in the MEGAN2.1 biogenic inventory in the default GEOS-Chem simulations
causing the overestimation of the HCHO changes in response to droughts.
The MOFLUX site provides the only long-term ground-based isoprene flux observations covering multiple drought
severities. We developed a drought stress algorithm based on the MOFLUX site following Jiang et al. (2018), and the
algorithm improves the HCHO simulation at the MOFLUX grid while underestimating HCHO after all the SE US
grids are included. By comparison, the OMI-based drought stress algorithm derived from the different HCHO-
temperature sensitivities between OMI and GEOS-Chem can reflect better spatial coverage and nearly removes the
positive bias between OMI and the default simulations seen from a test simulation in May-September 2012 over the
SE US.
The long-term simulation with the OMI-based drought stress factor can significantly reduce the biogenic isoprene
emissions by $0.35\times10^{-10}$ kg m$^{-2}$ s$^{-1}$ (14.24%), $0.43\times10^{-10}$ kg m$^{-2}$ s$^{-1}$ (16.57%), $0.49\times10^{-10}$ kg m$^{-2}$ s$^{-1}$ (17.49%), $0.58\times10^{-10}$
kg m$^{-2}$ s$^{-1}$ (18.66%) and $0.65\times10^{-10}$ kg m$^{-2}$ s$^{-1}$ (20.74%) from D0 to D4, respectively, which consequently leads to a
better agreement between OMI and simulated HCHO column. Despite lowering emissions relative to the no-stress
simulation, OMI_Stress_GC simulates a non-uniform trend of increasing isoprene emissions with drought severity
that is consistent with OMI HCHO and MOFLUX. Relative to N0, the simulated increase in isoprene emissions is 15-
18% under D0-D1, increasing to 26% at D2 and peaking at 37% at D3, followed by a slight decrease to 35% at D4.
The observed MDA8 O$_3$ and OA over the SE US show a similar increase pattern with HCHO. The OMI-based drought
stress algorithm also helps reduce the mean bias of OA by 0.30 μg/m$^3$ (6.60%), 0.46 μg/m$^3$ (8.98%) 0.60 μg/m$^3$
(10.07%), 0.67 μg/m$^3$ (10.85%), 0.62 μg/m$^3$ (10.88%), 0.74 μg/m$^3$ (11.71%) from N0 to D4 over the SE US compared
with the high positive bias of more than 2.02 μg/m$^3$ (50.83%) without the drought stress. By contrast, the MDA8 O$_3$
response to the reduced biogenic isoprene caused by the drought stress factor presents a spatial disparity due to the
nonlinear O$_3$ chemistry. Places with high isoprene emissions show an increase of MDA8 O$_3$ by less than 1 ppbv, which
slightly improves the simulated drought-O$_3$ relationship. For the regions with low isoprene emissions in the SE US,
the drought stress factor reduces MDA8 O$_3$ by 1-3 ppbv.
This study reveals an increasingly higher level of biogenic isoprene under drought conditions over the regions with
high vegetation coverage. As drought is predicted to become more frequent in a warming climate (Cook et al., 2018),
it is essential to update current biogenic emission inventories by adding a drought stress factor and to improve the
constraints of isoprene chemistry in the climate chemistry models in order to have a better projection of air quality in
the future. We demonstrate the feasibility of applying satellite data to the development of drought stress algorithms
when ground-based measurements are limited. Our attempt here is a top-down approach and used temperature as the
only parameter to adjust isoprene emissions under drought conditions. The water stress threshold in our algorithm is
used only as a triggering parameter; that is, it is used to determine whether a grid is in drought or not and thus can be
replaced with other drought-identifying approaches. One issue with our approach is the type of temperature data to be
used in the algorithm. Ideally, it should be leaf temperature because this is what regulates stomata at the process level.
However, leaf temperature is not readily available from meteorological fields that drive CTMs. MEGAN uses 2 m air
temperature to parameterize isoprene emissions, and thus our algorithm uses the same temperature. More biogenic
emission flux observations covering different vegetation types and drought severities will be helpful to better depict
the relationships between biogenic VOCs and drought stress.
**Acknowledgment**
This work was supported by NASA Atmospheric Composition Modeling and Analysis Program (80NSSC19K0986).
The development of the ecophysiology module in GEOS-Chem has also been supported by the General Research Fund
(14306220) granted by the Hong Kong Research Grants Council. The authors thank NASA Langley Research Center
for the OMI HCHO column data and the National Drought Mitigation Center for making and providing the USDM
maps. Roger Seco was supported by grants RYC2020-029216-I and CEX2018-000794-S funded by
MCIN/AEI/10.13039/501100011033 and by the European Social Fund "ESF Investing in your future".
**Data Availability**
GEOS-Chem model is publicly available at http://www.geos-chem.org (Bey et al., 2001). USDM shapefiles are
download from https://droughtmonitor.unl.edu/DmData/GISData.aspx (Svoboda et al., 2002). LAI is obtained from
http://geoschemdata.wustl.edu/ExtData/HEMCO/Yuan_XLAI/v2021-06/ (Yuan et al., 2011). $O_3$ and organic carbon
observational data can be downloaded via https://aqs.epa.gov/aqsweb/documents/data_mart_welcome.html (Schnell
et al., 2014). Observational isoprene measurements at MOFLUX are from Potosnak et al. 2014 and Seco et al. 2015
and are available upon request from co-author Alex Guenther. OMI Satellite HCHO and $NO_2$ columns are available
publicly at https://cmr.earthdata.nasa.gov/search/concepts/C1626121562-GES_DISC.html (Chance, 2019) and
https://disc.gsfc.nasa.gov/datasets/OMNO2d_003/summary (Nickolay et al., 2019), respectively.
**Competing interests**
The authors declare that they have no conflict of interest.
**Author contributions**
YW conceived the research idea. NL and WL conducted the model simulation and data analysis. JCYL and APKT
created the ecophysiology module. AG, MJP and RS provided the field observations. All authors contributed to the
interpretation of the results and the preparation of the manuscript

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
