# Peer review of "Satellite-derived Constraints on the Effect of Drought Stress on Biogenic Isoprene Emissions in the Southeast US"

_EGUsphere, 2022_

## Author Comment (AC1)

**Reply to Reviewers**

We sincerely appreciate all the reviewers for their constructive comments to improve the manuscript. Their comments are reproduced below followed by our responses in blue. The corresponding edits in the manuscript are highlighted with track changes.

**Reviewer #1:**
General Comments:
This study investigates the impacts of drought conditions on isoprene emissions. By combining observational data including satellite product and model simulations, the authors aim to improve the quantification of biogenic isoprene emissions in response to drought stress. This is a very important topic and the results are of great importance to the community. The paper is in general well written and I only have a few relatively minor comments.

Specific Comments:
(1) In using the HCHO data to derive the changes in isoprene emissions (drought vs no-drought conditions), I believe the underlying assumption is that there is no significant changes in other factors (e.g. chemistry, emissions of ozone precursors such as NOx from soil) during drought conditions – this is likely true, but it would be very helpful to point out and discuss in the text.
**Response**: Thank you for raising this important point. The constraints on the changes in isoprene emissions between drought and non-drought conditions were based on the difference between modeled and OMI HCHO columns, so we replied on the GEOS-Chem model to simulate the changes in other factors (e.g., chemistry, meteorology, emissions including NOx from soil). Therefore, the underlying assumption is that the GEOS-Chem model has no significant bias in predicting HCHO columns due to factors other than isoprene emissions between drought and non-drought conditions. The assumption is reasonable because the GEOS-Chem model uses reanalysis meteorology, state-of-the-science isoprene oxidation schemes, time-specific anthropogenic emissions and fire emissions, and natural emissions calculated online using model meteorology. Also, the model has been evaluated extensively against observations in the study region. We added the discussion on the assumption in Line 311 – 317.

In light of this comment and another comment from Reviewer #2, we did recognize one bias in the model that could potentially affect our analysis, which is the model overestimation of anthropogenic NOx emissions in the Southeast US when using the NEI emissions (e.g., Travis et al., 2016; Kaiser et al., 2018). As the HCHO formation from isoprene depends on $NO_x$, a bias in model $NO_x$ could affect the yield of HCHO from isoprene emissions, thereby influencing the difference between modeled and OMI HCHO column. To investigate this issue, we first compared tropospheric $NO_2$ columns between OMI and the default GEOS-Chem (e.g., without isoprene drought stress or NoStress_GC) in Figure R1. Indeed, we found NoStress_GC overestimates OMI $NO_2$ columns by ~42% for the SE US overall (solid lines in Figure R1), which is consistent with previous studies showing the default NEI2011 NOx emissions were biased high (Travis et al., 2016; Kaiser et al., 2018). Nevertheless, we found the bias in model $NO_2$ column was nearly constant from N0 (non-drought) to D4 (extreme drought), and model predicted $NO_2$ column changes under D0-D4 drought relative to non-drought (N0) were within 10% of what OMI observed (dashed lines in Figure R1; note the different scales from solid lines). Note $NO_2$ columns also show an increasing pattern from N0 to D4 yet with a much

smaller extent (less than 9%) than HCHO, which is consistent with our previous study (Li et al., 2022). The comparison shows that the model predicts the changes in $NO_x$ emissions with USDM very well despite the high bias in anthropogenic NOx emissions. This also implies that the changes in the natural sources of $NO_x$ (e.g., biomass burning and soil $NO_x$) with droughts are well represented by NoStress_GC.

[Figure]

**Figure R1**: Tropospheric $NO_2$ columns averaged by USDM levels from OMI and NoStress_GC sampled during the midday hours (13:30 LT; solid lines), and their respective changes from non-drought (N0) conditions (dashed lines). Note the different units between the solid and dashed lines shown in the legend. The calculation is based on the grids with the presence of all USDM levels.

Figure R1 was added to the manuscript as the new Figure 4b with associated discussion inserted in Line 236-251.

(2) Thinking about the future development of biogenic emission schemes used in CTMs, such as MEGAN, do you think it's easier to apply a drought stress factor as done here or simply add the precipitation (probably precipitation history over a certain period) in the MEGAN parameterization? I feel each has their own advantage. e.g., precipitation data is already there, just like other metfields like temperature, so you can easily do the calculations on the fly. Some discussion on this would be particularly helpful for the modeling community.

**Response**: Although the met fields (e.g., precipitation and temperature) are readily available, it is not straightforward to directly link them with the water stress vegetations are experiencing during droughts because of the complex canopy structures and root systems and the fact that different types of plant function types may have different water stress threshold. This is why soil moisture was used in the initial MEGAN framework to represent drought stress (Guenther et al., 2006) but this initial framework was not widely adopted due to issues with soil moisture database and wilting point threshold as discussed in the manuscript. Our attempt here took a top-down approach and used temperature as the only parameter to adjust isoprene emissions under drought conditions. The water stress threshold in our algorithm is used only as a triggering parameter; that is, it is used to determine a grid is in drought or not. One issue with our approach is the type of temperature data to be used in the algorithm. Ideally it should be leaf temperature because that is what regulates stomata at the process level. However, leaf temperature is not readily available from met fields. MEGAN uses 2 m air temperature to parameterize isoprene

emissions as a function of temperature, and thus our algorithm uses the same temperature. When more field observations focusing on isoprene emissions under droughts become available, a more robust parameterization of drought stress as a function of met fields may be developed as a function of different plant function types. Related discussions are added to Line 578-584.

**Reviewer #2**
General Comments:
This study employs the GEOS-Chem CTM (driven by MEGANv2.1 biogenic emissions) along with OMI HCHO measurements and ground-based flux measurements to derive constraints on the isoprene emission response to drought stress in the southeast US. The authors then implement an updated drought stress parameterization in MEGAN to investigate the impacts of drought-associated isoprene emission reductions (as compared to the baseline MEGAN implementation) on air quality.

I found this paper to be interesting and generally well-written; it is a nice application of a long-term satellite data record to advance our understanding of isoprene emission processes and improve our ability to simulate those processes in models. However, I think the paper needs more discussion and evaluation of the uncertainties associated with NOx biases in the GEOS-Chem simulations used here, and how they might impact interpretation of the results. I also think additional space should be devoted to more explanation of the potential impacts of the bias correction applied to the OMI HCHO data, and the other data adjustments that are performed. Specific recommendations are listed below. After these revisions I would recommend publication.

Specific Comments:
(1) Line 45-47: Travis et al. (2016) showed that NEI2011 emissions are biased high in the SE US, and Kaiser et al. (2018) thus applied a 60% reduction in NEI2011 anthropogenic NOx sources (other than power plants) to account for this in their OMI HCHO-based optimization of isoprene emissions over the region. Have the authors applied similar NOx adjustments in their simulations here?
**Response**: We did not apply such adjustments in the simulation and used the default NEI2011 $NO_x$ emissions. To test whether this high NOx emission bias can potentially affect our conclusions, we did a sensitivity simulation of reducing the NEI2011 $NO_x$ emissions by 50% over the SE US in JJA 2011-2013, which covers both non drought and severe drought periods. For example, most of the SE US was under droughts during the summertime of 2011-2012, while 2013 was a less drought-stricken year (Figure 1 in the main text). Figure R2a-b shows the results of HCHO columns and their changes from the sensitivity test. The overall reduction of HCHO columns caused by the halved NOx emissions is small and nearly constant among USDM levels, ranging from -0.044×$10^{16}$ molec cm$^{-2}$ (2.6%) to -0.056×$10^{16}$ molec cm$^{-2}$ (3.5%). This indicates that the high NEI2011 $NO_x$ emission is not the main reason for the overestimation of HCHO columns in NoStress_GC under droughts, and we thus conclude that it does not have significant impact on our results. Figure R2 was added to the supplement file as Figure S1 with associated discussions inserted in Line 236-251.

[Figure]

**Figure R2**: (a) No_Stress_GC simulated HCHO columns in JJA 2011-2013 at each USDM level. (b-d) Sensitivity tests of the HCHO column changes when (b) NEI2011 NO$_x$ emissions are reduced by 50% in the SE US, (c) GFED4 wildfire emissions are turned off, and (d) keep PBL in 2011-2012 (drought years) the same as in 2013 (non-drought year). The numbers in each panel indicate the regional mean values.

(2) Line 127: Do the authors have thoughts as to what type of uncertainty is introduced by using a single bias correction factor for the long-term OMI HCHO record used here? The 1.59 factor was derived by Zhu et al. (2016) with respect to aircraft measurements taken in a specific summer (2013), however, I wonder if temporally-varying biases are possible given that the HCHO background likely changes with time.
**Response**: The factor of 1.59 suggested by Zhu et al. (2016) works well in matching HCHO columns between OMI and NoStress_GC daily outputs factored by 1.12 at non-drought conditions (N0). However, after we changed to use the midday (13:30 LT) outputs as suggested by the next comment below (specific comment #3), the 1.59 factor makes OMI HCHO columns ~ 8.6% higher than the model at N0. Therefore, it is possible that this factor is temporally varying. To achieve an unbiased simulation at N0, we decided to apply 1.5, instead of 1.59, on

OMI data to correct the bias, which is the correction factor used in the long-term analysis of OMI HCHO columns by Shen et al. (2019).

(3) Line 149: Why not just sample the model at approximately the time of the OMI overpass? It seems like this would be more robust than scaling the daily mean data by a single conversion factor everywhere and at all times. Can the authors discuss potential uncertainty associated with this assumption?

**Response**: The initial analysis was done using the daily outputs at hand. To examine the possible bias, we reran the model and sampled the HCHO columns of NoStress_GC during the midday hours (13:30 LT) to match with the overpass time of OMI. There were minor changes in numerical values in the manuscript due to this change, but the overall conclusion is not affected. We re-calculated the formula in Table 1 using the midday data and found only trivial changes in the fitting. There was a slight change (less than 1%) in the drought stress algorithm (Equation 4 in main texts), from $\frac{\Omega_{OMI}}{\Omega_{GC}} = 382\, e^{-0.02T}\ (\beta_t < 0.6, T > 300\,K)$ in the original manuscript to $\frac{\Omega_{OMI}}{\Omega_{GC}} = 380.10 e^{-0.02T}\ (\beta_t < 0.6, T > 300\,K)$. Therefore, we did not rerun the OMI_Stress_GC simulation as insignificant changes are expected. We updated the related figures, Table 1, and discussions throughout the manuscript.

(4) Line 205-207: While I agree with the authors that isoprene is probably the dominant "missing process", is there any literature that discusses changes in other factors (e.g., biomass burning, mixing height, etc.) during SE US drought that the authors can point to here? I see in Fig. 4 that they ruled out changes in anthropogenic VOCs, which is a helpful addition.

**Response**: Thanks for the suggestions. While we did not find any literature systematically evaluating the HCHO simulation in response to wildfire emissions under drought conditions, there are existing case studies which focused on several filed campaigns and reported an underestimation of HCHO levels near the fire plumes partly due to the insufficient hydrocarbon emissions and the underrepresented plume chemistry (Alvarado et al., 2020; Liao et al., 2021; Zhao et al., 2022). Based on these, we suspect biomass burning is an unlike factor causing the model to overestimate HCHO columns under drought. A deeper planetary boundary layer (PBL) is expected under droughts primarily due to a larger sensible height flux released from dry soil (Miralles et al., 2014). Indeed, the MERRA-2 PBL height used in our simulation increases by 12.42%, 17.79%, 20.99%, 26.21%, and 29.52% from D0 to D4 relative to the value of 1589 m at N0 in the SE US during midday (13:30 LT). Considering the PBL heights in MERRA-2 agree well with observations with only an overall 200 m low bias (Guo et al., 2021), we suppose mixing heights should not be able to cause such a high bias of HCHO column. To further quantify the effects of these two processes on the changes of HCHO columns between drought and non-drought conditions, we conducted two additional sensitivity tests: (1) turning off wildfire emission inventory (GFED4) during 2011-2013 JJA and (2) keeping PBL constant in 2013 (normal year) for 2011-2012 (drought years) JJA. The results in Figure R2c-d show trivial changes in the simulated HCHO columns, which verifies our assumptions mentioned above. These discussions were added to Line 257-269.

(5) Lines 219-221: Could domain-mean temperature also be added to Fig. 4? If the more severe drought time periods are also warmer, then the increase in isoprene emissions is not really

surprising despite the LAI reductions, given the strong exponential dependence of those emissions on temperature.

**Response**: This is a good suggestion. The domain-mean surface (2m) temperature during midday (13:30 LT) indeed shows an increasing pattern with a value of 304.29K, 305.80K, 306.40K, 306.96K, 307.68K, 308.23K from N0 to D4, respectively. However, after scaling by the value at N0 in the same way as other variables in Figure 4, the temperature changes are around 1% and thus not noticeable. Thus, instead of adding a new line to Figure 4 which is not legible, we added some texts on the temperature changes after the discussion of LAI changes in Line 229-230.

(6) Line 224-228: I don't typically think of the SE US as a low NOx environment; however, even if it were, I think the buffered response is misrepresented as described here. It simply reflects the fact that HCHO is less sensitive to OH variability because it's loss to photolysis still occurs at low OH. However, the HCHO yield from isoprene varies as a function of NOx (and, thus, OH), so any NOx bias in the model can lead to an HCHO bias. Even after their NOx emission adjustments as discussed above, Kaiser et al. (2018) demonstrated that spatially-varying NOx biases lead to biases in the modeled HCHO column in the SE US. Have the authors evaluated the model NOx in their study region? I think a comparison to OMI NO2 would be a very nice addition to this paper, and would strengthen the argument that the overestimate of HCHO in the model is due to emissions and not a bias in the formation rate of HCHO.

**Response**: The reviewer's point is well taken. We revised the text and stated HCHO photolysis under low OH was the main reason for the buffered response of HCHO to changes in isoprene emissions (Line 231-234).

We agree with the reviewer's point regarding the model bias in NOx emissions. As suggested, we compared tropospheric $NO_2$ tropospheric columns from OMI and NoStress_GC in Figure R3. Using the default NEI2011 $NO_x$ emissions without any adjustment, NoStress_GC indeed overestimates $NO_2$ columns by ~42% for the SE US overall (solid lines in Figure R3). Since HEMCO (Harmonized Emissions Component) in GEOS-Chem applies year-specific factors to scale NEI2011 emissions to other years, the high $NO_x$ emissions in 2011 were propagated to other years and that is why the model constantly has a high bias in $NO_2$ columns across the USDM levels. It is noteworthy that this bias is nearly constant from N0 to D4, making almost the same $NO_2$ column changes relative to non-drought (N0) conditions between OMI and NoStress_GC (dashed lines in Figure R3; note the different scales from solid lines). This rules out the possibility that the high $NO_x$ bias in the model is the reason to explain the model overestimation of HCHO columns under drought conditions. Despite the high bias in $NO_2$ column, the model captures the changes in $NO_2$ column with USDM very well, which indicates the model well represents the change in the natural sources of $NO_x$ (e.g., biomass burning and soil $NO_x$) with droughts. Figure R3 was added to the original Figure 4 as a subplot with associated discussion inserted in Line 236-251.

[Figure]

**Figure R3**: Tropospheric $NO_2$ columns averaged by USDM levels from OMI and NoStress_GC sampled during the midday hours (13;30 LT; solid lines), and their respective changes from non-drought (N0) conditions (dashed lines). Note the different scales between the solid and dashed lines. The calculation is based on the grids with the presence of all USDM levels.

(7) Line 302: The discussion here is confusing—the authors talk about "downscaling" the GC emissions for comparison to MOFLUX, but what they've actually done is scale them up by 1.42, correct? It sounds like this number represents the mean MOFLUX-GC relative bias during N0?
**Response**: Correct. The factor was meant to eliminate the systematic bias during N0 and we attributed the bias to the point-to-grid comparison. We have rephrased the sentences to clarify.

(8) Figure 6b: I find it very difficult to see the direction of change between Nostress_GC and MOFLUX_stress_GC in this scatterplot. Suggest either having two panels or maybe adding the MOFLUX_stress_GC predictions as an additional line in Figure 6a.
**Response**: A new blue line representing MOFLUX_Stress_GC was added to Figure 6a.

(9) Lines 326-345 and Figure 7: I think these LAI-normalized HCHO vs temperature curves are a nice way to show the data, but I'm not clear as to why the model predicts an increasing temperature dependence as drought severity increases in the model. Isn't the temperature dependence a fixed function for each plant type in MEGAN? Does the variation reflect the temperature "memory" effect on emissions that presumably increases during drought (i.e. I think MEGAN actually accounts for the previous two weeks' temperatures or something along those lines?) or is it something else that's changing (such as an increase in clear, sunny days as the drought progresses, which would increase PAR)? Can the authors discuss this a bit here?
**Response**: Thanks for bringing up a good question and providing the potential answers. To verify whether these two possible processes are responsible for the overdependence of isoprene emissions on temperature under droughts, we compared the changes in 15-day mean surface temperature (PT_15) and photosynthetically active radiation (PAR) with midday surface temperature (T) at each USDM level over the SE US (Figure R4a-b). PT_15 is higher under droughts than non-drought conditions, especially at D3-D4, when the surface temperature is greater than 300 K, while the value of PAR is not. This indicates that temperature 'memory' effects are indeed enhanced under droughts as the reviewer suggested. We also examined the

temperature activity factor ($\gamma_T$) in Equation (1) for the calculation of the isoprene emission factor (main text) in Figure R4c. The value of $\gamma_T$ shows a higher exponential dependence on temperature under higher PT_15, which resembles the stronger relationships between HCHO/LAI and temperature under droughts as demonstrated in Figure 7 (main texts). This further verifies the higher PT_15 values under droughts are responsible for the overestimation of isoprene emissions under high temperatures. Figure R4 was added to the supplement file as Figure S4 with the related discussion inserted in Line 402-405.

[Figure]

**Figure R4**: Surface temperature (T) binned15-day mean surface temperature (PT_15; a) and photosynthetically active radiation (PAR; b) during midday hours (13:30 LT) at each USDM level from archived MERRA2 meteorology over the SE US. (c) Temperature factor ($\gamma_T$) in MEGAN2.1 changes with surface temperature under different PT_15 values.

(10) Lines 382-383: Can the authors discuss a bit more why they think the MOFLUX comparison doesn't do a good job representing drought stress dependence across the SE US? Is there something unique about the Ozarks ecosystem that doesn't apply across the region more generally?

**Response**: Although the Ozarks ecosystem is the so-called 'isoprene-volcano' region in the SE US, the site has relatively thin soils, which prevents the trees from absorbing moisture from deep soil layers and is thus vulnerable to water stress (Opacka et al., 2022). Isoprene emissions measured here are therefore more sensitive to droughts and the same extent of drought stress is likely too strong to be applied to other regions. We add such explanations in Line 430-433.

(11) Figure 10: I wonder if it would make more sense for the third column to be OMI_Stress_GC – OMI HCHO to demonstrate the improved agreement?

**Response**: We have considered using the suggested value when writing the manuscript, but finally chose the OMI_Stress_GC – NoStress_GC column because it provides a direct display of how the application of the OMI-derived drought stress factor can change the model results. The improved agreement was mentioned in the main texts and can be easily derived from the mean

biases listed in each panel of the second and third columns. We further created a Figure R5 (added as Figure S5) showing the differences between OMI_Stress_GC simulated and observed HCHO columns, MDA8 ozone, and OA.

[Figure]

**Figure R5**: Mean differences between OMI_Stress_GC simulated and observed HCHO columns (a), MDA8 ozone (b), and OA (c). Numbers at the bottom-left corner of each panel indicate the SE US (black box) regional mean bias (MB) and root mean square error (RMSE).

(12) Section 5: These results are interesting and a nice addition to the paper, but I'm not sure they demonstrate applicability over regions outside the SE US. Consider perhaps restricting the comparisons just to that region? Also, I think a discussion of how the authors deal with NOx biases in the model would help in the interpretation of these results.
**Response**: Since Figure 3 has demonstrated isoprene emissions and HCHO column changes in the SE US, using SE US only would contain repeated information. Expanding the maps to the contiguous US can provide additional information on whether the drought stress algorithm can impose large changes on other US regions although it is derived based on the SE US data only. We pointed out the caveat in Line 444 that the resulted changes from the drought stress outside of SE US need to be interpreted with caution. The $NO_x$ biases in the model have been addressed above.

Technical Comments:
(1) Line 27: By "non-uniform" I think the authors mean "non-linear"?
**Response**: Correct. We have changed it to 'non-linear'.
(2) Line 28: "trend of increase" is awkward. Do you mean "increasing trend"?
**Response**: Thanks for the suggestion. It has been changed accordingly.
(3) Line 67: Consider editing the beginning of this sentence to "With wide spatiotemporal coverage, satellites provide…"
**Response**: Thanks and done.
(4) Line 72: I would add "e.g." before the citations here, as this is far from an exhaustive list.
**Response**: Done.
(5) Line 79: Change "the monthly" to "a monthly"
**Response**: Done.
(6) Figure 2: I would label the panel titles and the axes with the same names used in the text (GCHCHO_Nostress and OMHCHOd). Also, the units should be denoted as "molec cm-2" instead of "mole cm-2" for this and all other figures.
**Response**: Done.
(7) Figure 3: I would label the GEOS-Chem panels (b and c) as "Nostress_GC" since this is how it is denoted in the text.
**Response**: Done.
(8) Figure 4: Label the curves corresponding to GEOS-Chem as "Nostress_GC" to avoid confusion.
**Response**: Done.
(9) Line 147: Change "simulating" to "simulated"
**Response**: Done.
(10) Line 170: "regions" should be "region"
**Response**: Done.
(11) Line 193-194: This sentence is awkward. Consider revising.
**Response**: Done.
(12) Figure 7: Label the curves corresponding to GC as "Nostress_GC"
**Response**: Done.
(13) Line 331: "sensitives" should be "sensitivities"
**Response**: Done.

**Community Comments**

This manuscript provides an account of the impacts of drought on isoprene emissions using satellite measurements combined with model simulations and ground-based observations. As noted by the authors in the Introduction, efforts in establishing the effects of drought on isoprene emissions are limited by the scarcity of observations currently available. It would therefore be worth adding to the Introduction recent work from the WIsDOM field campaign in the UK to provide up-to-date context to the readers. In particular:
**Response**: As suggested, we pointed out this filed work in Line 65-59 of the Introduction. The increased isoprene concentrations during this field campaign support the isoprene emission changes at the MOFLUX site under mild droughts.

(1) Lines 48-49: The work of Otu-Larbi et al. (GBC, 2020) used observations from the WIsDOM site combined with a canopy model to show how not including a drought stress factor in the emission algorithm led to severe underestimates of the observed isoprene concentrations.
**Response**: The citation of this work was added.

(2) Lines 62-66: Similarly, the work by Ferracci el al. (GRL, 2020) supports the conclusions from the MOFLUX studies by observing a similar behaviour in a mid-latitude temperate forest in the UK where prolonged drought is rare. This is, to date, the only dataset of ecosystem-scale observations of isoprene during drought other than the MOFLUX study.
**Response**: Thanks for pointing this out. This work is quite related to the topic and was added to the references.

[revised manuscript text omitted]

---

## Author Response (AR2)

**Reply to Reviewers**

We sincerely appreciate the two reviewers for their time and efforts to improve the revised manuscript. The minor comments from the second reviewer are reproduced below followed by our responses in blue. The corresponding edits in the manuscript are highlighted with track changes.

**Reviewer #2:**

The added analysis of additional impacts that affect HCHO abundance and that may vary with drought severity (such as NOx and mixing height) has strengthened this paper, and help provide additional support for its conclusions on drought impacts on isoprene emissions. I have a few minor technical corrections to suggest, but would otherwise recommend publication.

(1) Line 137: Suggest including a reference for the OMI NO2 product used here.

**Response**: Thanks for the suggestion. The reference (Nickolay et al., 2019) was added in the main texts.

(2) Figure 4b: I think the units for the dashed lines are incorrect (should be $10^{14}$ molec/cm$^2$?). Also, the title of the panel suggests that the dashed lines represent relative changes in NO2, but they are absolute differences from N0, correct?

**Response**: Thanks for catching the error. The unit has been changed to $10^{14}$ molec/cm$^2$. It is correct that the dashed lines are absolute differences from N0. We have clarified this in the subtitle of Figure4b in the main texts.

(3) Line 231-232: While it makes sense that one might expect more HCHO photolysis during drought due to more clear, sunny days, are there any studies to cite here that have documented this effect?

**Response**: Two references (Naimark et al., 2021; Wang et al., 2017) were added to the main texts.

**References:**

Naimark, J. G., Fiore, A. M., Jin, X., Wang, Y., Klovenski, E., and Braneon, C.: Evaluating Drought Responses of Surface Ozone Precursor Proxies: Variations With Land Cover Type, Precipitation, and Temperature, Geophys. Res. Lett., 48, e2020GL091520, https://doi.org/10.1029/2020GL091520, 2021.

Nickolay A. Krotkov, Lok N. Lamsal, Sergey V. Marchenko, Edward A. Celarier, Eric J.Bucsela, William H. Swartz, Joanna Joiner and the OMI core team (2019), OMI/Aura NO2 Cloud-Screened Total and Tropospheric Column L3 Global Gridded 0.25 degree x 0.25 degree V3, NASA Goddard Space Flight Center, Goddard Earth Sciences Data and Information Services Center (GES DISC), Accessed: [last access: 4 October 2022], https://doi.org/10.5067/Aura/OMI/DATA3007.

Wang, Y., Xie, Y., Dong, W., Ming, Y., Wang, J., and Shen, L.: Adverse effects of increasing drought on air quality via natural processes, Atmospheric Chem. Phys., 17, 12827–12843, https://doi.org/10.5194/acp-17-12827-2017, 2017.